

SciPost Phys. 1(1), 006 (2016)

# Dispersive hydrodynamics of nonlinear polarization waves in two-component Bose-Einstein condensates

T. Congy[1*], A. M. Kamchatnov[2] and N. Pavloff[1]

**1** LPTMS, CNRS, Univ. Paris-Sud, Université Paris-Saclay, 91405 Orsay, France
**2** Institute of Spectroscopy, Russian Academy of Sciences, Troitsk, Moscow, 108840, Russia

\* thibault.congy@u-psud.fr

## Abstract

We study one dimensional mixtures of two-component Bose-Einstein condensates in the limit where the intra-species and inter-species interaction constants are very close. Near the mixing-demixing transition the polarization and the density dynamics decouple. We study the nonlinear polarization waves, show that they obey a universal (i.e., parameter free) dynamical description, identify a new type of algebraic soliton, explicitly write simple wave solutions, and study the Gurevich-Pitaevskii problem in this context.

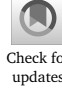

# 1  Introduction

As demonstrated in various physical contexts, the interplay between dispersive and nonlinear effects can lead to a number of spectacular phenomena as, for instance, the formation of solitons and vortices. Bose-Einstein condensates (BECs) display both effects: (i) dispersion which is due to the so-called quantum pressure and (ii) nonlinear properties due to the interaction between the condensed atoms. Already in a pioneering paper, Bogoliubov [1] showed that the combination of these two features yields reconstruction of the ground state of the many-particle system, with formation of new types of elementary excitations—Bogoliubov quasiparticles. The generalization of the Bogoliubov method to nonuniform time-dependent systems by Gross [2] and Pitaevskii [3] permitted to develop the theory of quantum vortices and later Tsuzuki [4] demonstrated the existence of dark solitons in a one dimensional model of weakly interacting bosons. After the experimental realization of BEC in ultracold gases, dark solitons were observed first in a one-dimensional geometry under the form of dips propagating along a stationary background [5, 6] and then in two dimensions under the form of stationary oblique solitons [7–9] generated by the flow of an exciton-polariton condensate past an obstacle [10, 11]. More complicated nonlinear wave structures were experimentally observed [12] and interpreted as dispersive shock waves, the description of which can be developed in the framework of Whitham's theory of modulations of nonlinear waves [12, 13] (for a recent review on modulation theory of nonlinear waves see, e.g., Ref. [14]).

The experimental realization of condensates consisting of two (or more) species has opened the possibility of studying even richer dynamics triggered by the additional degree(s) of freedom consisting in the relative motion of the components. These are new modes that can interact with each other leading, in particular, to different types of solitons. For two-component systems these new modes can be visualized as pertaining to two types of waves: "density waves" with in-phase motion of the two components and "polarization waves" with counter-phase motion of the components. In the simplest situations these two types of excitations decouple: the first type does not involve relative motion of the components and the second type of waves does not affect the total density of the condensate. In the small amplitude limit these two types of waves and the distinction between density and polarization excitations were studied, e.g., in Ref. [15].

It has been recently noticed [16] that the polarization dynamics can be separated from the density dynamics even for the case of large amplitude waves if the difference between intra- and inter-species interaction constants is small, and this observation was applied to the theory of polarization solitons—which were denoted as "magnetic solitons". In the present paper we extend the method of Ref. [16] to the general case of polarization dynamics in two-component BECs with small difference between the nonlinear interaction constants. In section 2 we derive the general equations of the polarization dynamics. In section 3 we study their traveling wave solutions that include, as a limiting case, the soliton solution found in [16] and in Sec. 4 we study the dispersionless limit of the nonlinear polarization waves. This forms the basis for discussing in section 5 the evolution of initial discontinuities in the polarization distribution. We show that such discontinuities evolve into a wave structure consisting in a rarefaction wave separated from a dispersive shock wave by a plateau with constant polarization and relative flow velocity. The main characteristics of this structure are calculated with the use of Whitham theory and are shown to compare very well with the results of numerical simulations. The relevance of our results for experimental studies is discussed in section 6. Our conclusions are summarized in Sec. 7 and some technical aspects are detailed in Appendixes A and B.

## 2 Model and main equations

We consider a one-dimensional BEC system described by a two-component spinor order parameter $\Psi(x,t) = (\psi_\uparrow, \psi_\downarrow)^t$ (where the superscript $^t$ denotes the transposition) obeying the following coupled Gross-Pitaevskii equations

$$i\hbar\partial_t\psi_{\uparrow,\downarrow} + \frac{\hbar^2}{2m}\partial_x^2\psi_{\uparrow,\downarrow} - \left[g\,|\psi_{\uparrow,\downarrow}|^2 + (g-\delta g)\,|\psi_{\downarrow,\uparrow}|^2\right]\psi_{\uparrow,\downarrow} = 0, \tag{1}$$

In Eqs. (1), it has been assumed that the two intra-species non linear coefficients $g_{\uparrow\uparrow}$ and $g_{\downarrow\downarrow}$ have the same value, denoted as $g$. For instance, this is exactly realized in the mixture of the two hyperfine states $|F = 1, m_F = \pm 1\rangle$ of $^{23}$Na [19], and, to a good approximation, in the mixture of hyperfine states of $^{87}$Rb considered in Ref. [20] ($|F, m_F\rangle = |1, 1\rangle$ and $|2, 2\rangle$). The inter-species coefficient $g_{\uparrow\downarrow}$ is written as $g - \delta g$, and we assume that

$$0 < \delta g \ll g. \tag{2}$$

Both conditions are realized in the above presented cases of $^{23}$Na ($\delta g/g \simeq 0.07$) and $^{87}$Rb ($\delta g/g \simeq 0.01$). The left condition is the mean-field miscibility condition of the two species (see, e.g., Refs. [21, 22]). The right condition will be shown later to lead to important simplifications in the dynamics of the system.

We represent the spinor wave function as

$$\begin{pmatrix} \psi_\uparrow \\ \psi_\downarrow \end{pmatrix} = \sqrt{\rho}\, e^{i\Phi/2} \begin{pmatrix} \cos\frac{\theta}{2}\, e^{-i\phi/2} e^{-i\mu_\uparrow t/\hbar} \\ \sin\frac{\theta}{2}\, e^{i\phi/2} e^{-i\mu_\downarrow t/\hbar} \end{pmatrix}. \tag{3}$$

In this expression, $\rho(x,t)$ is the total density and $\theta(x,t)$ governs the linear densities of the two components: $\rho_\uparrow(x,t) = |\psi_\uparrow|^2$ and $\rho_\downarrow(x,t) = |\psi_\downarrow|^2$ (cf. Eqs. (18) in the case of a constant total density $\rho_0$). $\Phi(x,t)$ and $\phi(x,t)$ act as potentials for the velocity fields $v_\uparrow$ and $v_\downarrow$ of the two components. Namely

$$v_\uparrow(x,t) = \frac{\hbar}{2m}(\Phi_x - \phi_x), \quad v_\downarrow(x,t) = \frac{\hbar}{2m}(\Phi_x + \phi_x). \tag{4}$$

By means of the substitution (3) the Gross-Pitaevskii system (1) is cast into the form

$$\hbar\rho_t + \frac{\hbar^2}{2m}\left[\rho(\Phi_x - \phi_x\cos\theta)\right]_x = 0,$$

$$\hbar\Phi_t + \frac{\hbar^2}{2m}\left(\frac{\rho_x^2}{2\rho^2} - \frac{\rho_{xx}}{\rho}\right) - \frac{\hbar^2}{2m}\frac{\cot\theta}{2\rho}(\rho\,\theta_x)_x + \frac{\hbar^2}{4m}(\Phi_x^2 + \phi_x^2 + \theta_x^2) + (2g-\delta g)(\rho - \rho_0) = 0,$$

$$\hbar\theta_t + \frac{\hbar^2}{2m\rho}(\phi_x\,\rho\sin\theta)_x + \frac{\hbar^2}{2m}\Phi_x\theta_x = 0,$$

$$\hbar\phi_t - \frac{\hbar^2}{2m\rho\sin\theta}(\rho\,\theta_x)_x + \frac{\hbar^2}{2m}\Phi_x\phi_x - \delta g\,\rho\cos\theta = 0,$$

$$\tag{5}$$

where it is assumed that at equilibrium both components are at rest and both have the same uniform density. The total density is denoted as $\rho_0$. In this case the chemical potentials take the same value: $\mu_\uparrow = \mu_\downarrow = (g-\delta g)\rho_0/2$. As is known, in such a system there are two types of waves that can be called "density" and "polarization" waves. In the small amplitude and long wavelength limit the velocity of polarization waves that correspond to the (mainly) relative motion of the components is equal to

$$c_p = \sqrt{\frac{\rho_0\delta g}{2m}}. \tag{6}$$

In the limit (2) $c_p$ is very small compared to the long wavelength velocity $c_d$ of density waves $[mc_d^2 = \rho_0(g - \delta g/2)]$. Following Ref. [16], we introduce also the "polarization healing length"

$$\xi_p = \frac{\hbar}{\sqrt{2m\rho_0\delta g}}. \tag{7}$$

Then the characteristic time scale for the polarization dynamics can be measured in units of

$$T_p = \frac{\xi_p}{c_p} = \frac{\hbar}{\rho_0\delta g}. \tag{8}$$

$T_p$ and $\xi_p$ are much larger than the corresponding characteristic time and length associated with density waves, and for the study of the polarization nonlinear waves it is thus appropriate to pass to the non-dimensional variables

$$\zeta = \frac{x}{\xi_p}, \qquad \tau = \frac{t}{T_p}. \tag{9}$$

Then a very important consequence can be inferred from the second equation (5) that, in new non-dimensional variables, can be written as

$$\frac{\rho - \rho_0}{\rho_0} = \frac{\delta g}{2g} \cdot \left\{ \frac{\cot\theta}{\rho}(\rho\,\theta_\zeta)_\zeta - \Phi_\tau - \frac{\rho_\zeta^2}{2\rho^2} + \frac{\rho_{\zeta\zeta}}{\rho} - \frac{1}{2}(\Phi_\zeta^2 + \phi_\zeta^2 + \theta_\zeta^2) + \frac{\rho}{\rho_0} - 1 \right\}. \tag{10}$$

We see that for $\theta \sim 1$ at space and time scales of order (7) and (8), correspondingly, the right-hand side becomes small if $\delta g/g \ll 1$. In this case we can assume at the leading order that $\rho \approx \rho_0$, so that the polarization hydrodynamics is decoupled from the density dynamics. This important feature of the two-component BEC dynamics with a small difference of the inter and intra-nonlinear constants was first indicated in Ref. [16] for the case of polarization solitons. The appearance of the $\cot\theta$-function in the first term in the braces shows that the condition $\delta g/g \ll 1$ should be complemented by another condition: $\theta$ should not be too close to zero or $\pi$ so that the right-hand side of (10) remains small. Thus, in addition to (2), we assume also that

$$\max\{\theta, \pi - \theta\} \gg \frac{\delta g}{g}. \tag{11}$$

This condition implies that the densities $\rho_\uparrow$ or $\rho_\downarrow$ are not too close to $\rho_0$ or 0, cf. Eqs. (18).

If the conditions (2) and (11) are fulfilled, then the density and polarization dynamics are decoupled and we can study the polarization dynamics separately assuming that $\rho(x,t) = \rho_0 = \text{const}$ and disregarding the second equation in the system (5). This approximation greatly simplifies the other equations. The first one reduces to

$$(\Phi_\zeta - \phi_\zeta\cos\theta)_\zeta = 0. \tag{12}$$

If we choose to work in a reference frame in which there is no flux of the total density, Eq. (12) simplifies to

$$\Phi_\zeta = \phi_\zeta\cos\theta, \tag{13}$$

and $\Phi_\zeta$ can then be excluded from the remaining two equations. This yields the system

$$\theta_\tau + 2\theta_\zeta\,\phi_\zeta\cos\theta + \phi_{\zeta\zeta}\sin\theta = 0,$$
$$\phi_\tau - \cos\theta(1 - \phi_\zeta^2) - \frac{\theta_{\zeta\zeta}}{\sin\theta} = 0. \tag{14}$$

This closed system of nonlinear equations shows that, for time scales of order $T_p$ and length scales of order $\xi_p$, the polarization degree of freedom decouples from the density degree of

freedom, even in the nonlinear regime. All dimensional parameters have been scaled out from Eqs. (14) which thus correspond to a universal behavior of polarization waves. Note here that the relevant characteristic time (8) and length (7) have been identified for equal densities of both components (and will keep the same value throughout the paper), but the validity of the system (14) does not rely on this assumption: it describes the polarization dynamics in the limit (11), for a system verifying (2). In this case, we see from Eq. (10), that the ratio of the amplitude of density waves with respect to the one of polarization waves is roughly of order $\delta g/g$.

The system (14) can be derived from the Hamilton principle of extremal action [16] for a Lagrangian $\Lambda = \int \mathscr{L} \, d\zeta$ with a Lagrangian density

$$\mathscr{L} = \phi_\tau \cos\theta - \frac{1}{2}\left[\theta_\zeta^2 + (\phi_\zeta^2 - 1)\sin^2\theta\right]. \tag{15}$$

From this expression we can write the (correctly dimensioned) energy of the system under the form

$$E = \frac{1}{2}\rho_0^2 \, \delta g \, \xi_p \int d\zeta \, u(\zeta, \tau). \tag{16}$$

where

$$u = \phi_\tau \frac{\partial \mathscr{L}}{\partial \phi_\tau} + \theta_\tau \frac{\partial \mathscr{L}}{\partial \theta_\tau} - \mathscr{L} = \frac{1}{2}\left[\theta_\zeta^2 + (\phi_\zeta^2 - 1)\sin^2\theta\right] \tag{17}$$

is the energy density corresponding to the Lagrangian (15). This expression coincides with the energy of ferromagnetic bodies in dissipationless Landau-Lifshitz theory [17] with account of dispersion and uniaxial easy-plane anisotropy.

The system (14) can be cast into other forms that may be more convenient in some instances. In particular, the angle $\theta$ is related to the density of each component by the formulas

$$\rho_\uparrow = \frac{1}{2}\rho_0(1 + \cos\theta), \qquad \rho_\downarrow = \frac{1}{2}\rho_0(1 - \cos\theta), \tag{18}$$

hence

$$w \equiv \cos\theta = \frac{\rho_\uparrow - \rho_\downarrow}{\rho_0} \tag{19}$$

is the variable describing the variations of the relative density. On the other hand,

$$v \equiv \phi_\zeta = \frac{v_\downarrow - v_\uparrow}{2c_p} \tag{20}$$

represents the non-dimensional relative velocity of the components. In terms of the two variables $(w, v)$ which have clear physical meanings, the system (14) takes the form

$$w_\tau - [(1 - w^2)v]_\zeta = 0,$$
$$v_\tau - [(1 - v^2)w]_\zeta + \left[\frac{1}{\sqrt{1-w^2}}\left(\frac{w_\zeta}{\sqrt{1-w^2}}\right)_\zeta\right]_\zeta = 0. \tag{21}$$

This system coincides with the one-dimensional version of the system derived in the recent preprint [18] for hydrodynamic description of magnetization dynamics in ferromagnetic thin films.

For subsonic flows with velocities $|v| < 1$ we can introduce a variable $\sigma$ such that

$$v = \cos\sigma, \tag{22}$$

and then in terms of $(\theta, \sigma)$-variables the system of equations of the polarization dynamics reads

$$\theta_\tau + 2\cos\theta \cdot \cos\sigma \cdot \theta_\zeta - \sin\theta \cdot \sin\sigma \cdot \sigma_\zeta = 0,$$
$$\sigma_\tau + 2\cos\theta \cdot \cos\sigma \cdot \sigma_\zeta - \sin\theta \cdot \sin\sigma \cdot \theta_\zeta + \frac{1}{\sin\sigma}\left(\frac{\theta_{\zeta\zeta}}{\sin\theta}\right)_\zeta = 0. \tag{23}$$

The importance of distinguishing subsonic from supersonic flows—an essential assumption for being able to write the relation (22)—can be seen from the following observation: consider a stationary uniform background characterized by a relative density $w_0$ and a relative velocity $v_0$. Linear perturbations of the form

$$w = w_0 + w', \quad v = v_0 + v', \quad \text{where} \quad w'(\zeta, \tau), v'(\zeta, \tau) \propto \exp[i(k\zeta - \omega\tau)],$$

obey the following dispersion relation:

$$\omega = \left(2w_0 v_0 \pm \sqrt{(1-w_0^2)(1-v_0^2)+k^2}\right)k. \tag{24}$$

By definition we always have $|w_0| \leq 1$, however $v_0$ can have any value, and for $|v_0| > 1$ the frequency $\omega$ is complex for small enough wavevectors $k$. This implies a long wavelength instability of supersonic relative motions of two-component superfluids, more precisely for a background relative velocity $v_\downarrow - v_\uparrow$ larger than $2c_p$. This mechanism of instability has been first theoretically studied in Ref. [23], and the regime (2) we consider here corresponds to what is denoted as "strong coupling" in this reference.

We note here for future use that, for subsonic flows with $w_0 = \cos\theta_0$ and $v_0 = \cos\sigma_0$, the dispersion relation (24) can be written as

$$\omega = \left(2\cos\sigma_0\cos\theta_0 \pm \sqrt{\sin^2\sigma_0\sin^2\theta_0 + k^2}\right)k. \tag{25}$$

The long wave length behavior of the dispersion relations (24) and (25) is linear and corresponds to a velocity of sound in the laboratory frame

$$c = 2w_0 v_0 \pm \sqrt{(1-w_0^2)(1-v_0^2)} = 2\cos\sigma_0\cos\theta_0 \pm \sin\sigma_0\sin\theta_0. \tag{26}$$

For a uniform system in which both components have equal densities ($w_0 = 0$) and no relative velocity ($v_0 = 0$) one gets $c = \pm 1$, i.e., going back to dimensional quantities, the speed of the polarization sound is $c_p$ as expected.

## 3 Traveling waves and solitons of polarization

In this section we consider traveling wave for which the physical variables $\theta$ and $v$ depend on $\xi = \zeta - V\tau$ only, $V$ being the phase velocity of the wave. In the framework of the system (14) this corresponds in making the *ansatz* that the velocity potential $\phi(\zeta, \tau)$ and $\theta(\zeta, \tau)$ can be represented as

$$\phi(\zeta, \tau) = q\zeta + \tilde{\phi}(\xi), \quad \theta = \theta(\xi). \tag{27}$$

Substitution into the first equation of the system (14), multiplication by $\sin\theta$ and integration give at once

$$\tilde{\phi}_\xi = -q + V \cdot \frac{B - \cos\theta}{\sin^2\theta}, \tag{28}$$

where $B$ is an integration constant. Substituting this expression into the second equation of the system (14) gives after simple transformations the equation

$$\theta_{\xi\xi} = V^2 \cdot \frac{(B - \cos\theta)(B\cos\theta - 1)}{\sin^3\theta} - \sin\theta\cos\theta + Vq\sin\theta. \tag{29}$$

Multiplication by $\theta_\xi$ and integration yield the final equation for the variable $w = \cos\theta$:

$$w_\xi^2 = -Q(w), \quad \text{with} \quad Q(w) = w^4 - 2Vqw^3 + (C-1)w^2 + 2V(q-VB)w + V^2(1+B^2) - C, \tag{30}$$

where $C$ is an integration constant. The four parameters $V, q, B, C$ can be expressed in terms of the four zeroes $w_1 \le w_2 \le w_3 \le w_4$ of the polynomial

$$Q(w) = \prod_{i=1}^{4}(w - w_i) = w^4 - s_1 w^3 + s_2 w^2 - s_3 w + s_4, \tag{31}$$

where the $s_i$'s are standard symmetric functions of the zeroes $w_i$ [1]. In particular, we obtain

$$V = \pm\frac{1}{2}\left[Q(1) + Q(-1) + 2\sqrt{Q(1)Q(-1)}\right]^{1/2}, \tag{32}$$

and

$$q = \frac{s_1}{2V}. \tag{33}$$

The solution of Eq. (30) can be expressed in terms of Jacobi elliptic functions and, without going into well-known details (see, e.g., [24]), we shall present here the final results.

The variable $w$ can oscillate between two zeroes of the polynomial $Q(w)$ where $Q(w) \le 0$ provided these two zeroes are located in the interval $[-1, 1]$. There are two possibilities, labeled as (A) and (B) below.

(A) In the first case the periodic solution corresponds to oscillations of $w$ in the interval

$$w_1 \le w \le w_2. \tag{34}$$

The solution of Eq. (30) can be written as

$$\xi = \int_{w_1}^{w} \frac{dw}{\sqrt{(w-w_1)(w_2-w)(w_3-w)(w_4-w)}}. \tag{35}$$

To simplify the notations, we put in (35) (and in all subsequent similar equations) the integration constant $\xi_0$ equal to zero. A standard calculation yields

$$w = w_2 - \frac{(w_2 - w_1)\text{cn}^2(W, m)}{1 + \frac{w_2 - w_1}{w_4 - w_2}\text{sn}^2(W, m)}, \tag{36}$$

where

$$W = \sqrt{(w_3 - w_1)(w_4 - w_2)}\,\xi/2, \quad m = \frac{(w_4 - w_3)(w_2 - w_1)}{(w_4 - w_2)(w_3 - w_1)}, \tag{37}$$

cn and sn being Jacobi elliptic functions [25]. The wavelength is given by

$$L = \frac{4K(m)}{\sqrt{(w_3 - w_1)(w_4 - w_2)}}, \tag{38}$$

---

[1] $s_1 = \sum_i w_i$, $s_2 = \sum_{i \ne j} w_i w_j$, $s_3 = \sum_{i \ne j \ne k \ne i} w_i w_j w_k$ and $s_4 = \Pi_i w_i$.

where $K(m)$ is the complete elliptic integral of the first kind [25]. In the limit $w_3 \to w_2$ ($m \to 1$) the wavelength tends to infinity and the solution (36) transforms to a soliton

$$w = w_2 - \frac{w_2 - w_1}{\cosh^2 W + \frac{w_2 - w_1}{w_4 - w_2} \sinh^2 W}. \tag{39}$$

This is a "dark soliton" for the variable $w = \cos\theta$.

The limit $m \to 0$ can be reached in two ways.

(i) If $w_2 \to w_1$, then we get

$$w \cong w_2 - \frac{1}{2}(w_2 - w_1)\cos[k(\zeta - V\tau)], \quad \text{where} \quad k = \sqrt{(w_3 - w_1)(w_4 - w_1)}. \tag{40}$$

This is a small-amplitude limit describing propagation of a harmonic wave.

(ii) If $w_4 = w_3$ but $w_1 \neq w_2$, then we get a nonlinear wave represented in terms of trigonometric functions:

$$w = w_2 - \frac{(w_2 - w_1)\cos^2 W}{1 + \frac{w_2 - w_1}{w_3 - w_2}\sin^2 W}, \quad \text{where} \quad W = \sqrt{(w_3 - w_1)(w_3 - w_2)}\,\xi/2. \tag{41}$$

If we take the limit $w_2 - w_1 \ll w_3 - w_1$ in this solution, then we return to the small-amplitude limit (40) with $w_4 = w_3$. On the other hand, if we take here the limit $w_2 \to w_3 = w_4$, then the trigonometric functions in (41) have a small argument and can be approximated by the first terms of their series expansions. This yields a solution which we denote as an "algebraic soliton":

$$w = w_2 - \frac{w_2 - w_1}{1 + (w_2 - w_1)^2(\zeta - V\tau)^2/4}. \tag{42}$$

(B) In the second case the variable $w$ oscillates in the interval

$$w_3 \leq w \leq w_4 \tag{43}$$

so that instead of (35) we get

$$\xi = \int_{w_3}^{w} \frac{dw}{\sqrt{(w - w_1)(w - w_2)(w - w_3)(w_4 - w)}}. \tag{44}$$

Again, a standard calculation yields

$$w = w_3 + \frac{(w_4 - w_3)\mathrm{cn}^2(W, m)}{1 + \frac{w_4 - w_3}{w_3 - w_1}\mathrm{sn}^2(W, m)}. \tag{45}$$

with the same definitions for $W$, $m$, and $L$ as in Eqs. (37) and (38). In the soliton limit $w_3 \to w_2$ ($m \to 1$) we get

$$w = w_2 + \frac{w_4 - w_2}{\cosh^2 W + \frac{w_4 - w_2}{w_2 - w_1}\sinh^2 W}. \tag{46}$$

This is a "bright soliton" for the variable $w = \cos\theta$.

Again, the limit $m \to 0$ can be reached in two ways.

(i) If $w_4 \to w_3$, then we obtain a small-amplitude harmonic wave

$$w \cong w_3 + \frac{1}{2}(w_4 - w_3)\cos[k(\zeta - V\tau)], \quad \text{where} \quad k = \sqrt{(w_3 - w_1)(w_3 - w_2)}. \tag{47}$$

This is a small-amplitude limit describing a harmonic wave.

(ii) If $w_2 = w_1$, then we obtain another nonlinear trigonometric solution,

$$w = w_3 + \frac{(w_4 - w_3)\cos^2 W}{1 + \frac{w_4 - w_3}{w_3 - w_1}\sin^2 W}, \quad \text{where} \quad W = \sqrt{(w_3 - w_1)(w_4 - w_1)}\,\xi/2. \tag{48}$$

If we assume in this solution $w_4 - w_3 \ll w_4 - w_1$, then we reproduce the small-amplitude limit (47) with $w_2 = w_1$. On the other hand, in the limit $w_3 \to w_2 = w_1$ we obtain the algebraic soliton solution:

$$w = w_1 + \frac{w_4 - w_1}{1 + (w_4 - w_1)^2(\zeta - V\tau)^2/4}. \tag{49}$$

This ends the general presentation of the different solutions of Eq. (30).

It is now interesting to discuss in more detail the soliton solutions which play a special role in the description of dispersive shock waves (Sec. 5.2). The bright soliton solution (46) corresponds to an increased number of particles in the "up" component:

$$\Delta N_\uparrow = \int dx \,(\rho_\uparrow^{\text{sol}} - \rho_\uparrow^{(0)}), \tag{50}$$

where $\rho_\uparrow^{(0)} = \rho_0(1 + w_2)/2$ is the background density of the up component and $\rho_\uparrow^{\text{sol}}(\zeta, \tau) = \rho_0(1 + w)/2$, $w(\xi)$ being given by (46). One gets

$$\Delta N_\uparrow = 2\rho_0 \xi_p \arctan\sqrt{\frac{w_4 - w_2}{w_2 - w_1}}. \tag{51}$$

The soliton is characterized by the three zeros $w_1$, $w_2(= w_3)$ and $w_4$ which relate to the physical variables $w_0$ (relative background density of the components), $V$ (velocity of the soliton) and $v_0$ (relative background velocity of the components) through

$$w_2 = w_0, \quad \text{and} \quad w_{4/1} = v_0(V - v_0 w_0) \pm \sqrt{(1 - v_0^2)[1 - (V - v_0 w_0)^2]}. \tag{52}$$

The energy of the soliton is the difference between the energy (16) of the system in the presence and in the absence of the soliton. It reads $E_{\text{sol}} = \frac{1}{2}\delta g \rho_0^2 \xi_p \mathcal{E} = \frac{1}{2}\hbar\rho_0 c_p \mathcal{E}$ where

$$\mathcal{E} = \int d\zeta\left[u(\xi) - \tfrac{1}{2}(v_0^2 - 1)(1 - w_0^2)\right], \tag{53}$$

$u(\xi)$ being here the energy density (17) computed for the distribution (46). It is shown in Appendix A that

$$\mathcal{E} = 2\sqrt{(w_4 - w_2)(w_2 - w_1)} = 2\sqrt{(1 - v_0^2)(1 - w_0^2) - (V - 2v_0 w_0)^2}. \tag{54}$$

The soliton solution found in [16] is reproduced from Eqs. (39) and (46) if we consider the situation where the two components have equal background densities ($w_0 = 0$), and no relative velocity ($v_0 = 0$). In this case, one gets from Eq. (52)

$$w_2 = w_3 = 0, \quad \text{and} \quad w_{4/1} = \pm\sqrt{1 - V^2}, \tag{55}$$

that is $Q(w) = w^2(w^2 - 1 + V^2)$ which agrees with formula (32). As a result we obtain

$$w = \cos\theta = \pm\frac{\sqrt{1 - V^2}}{\cosh\left[\sqrt{1 - V^2}(\zeta - V\tau)\right]}, \tag{56}$$

and Eqs. (18) give the corresponding densities of each component. From (51) and (54), one sees that this soliton corresponds to an increase of the number particles of the up component

$\Delta N_\uparrow = \frac{\pi}{2}\rho_0 \xi_p$ and to an energy $E_{\text{sol}} = \hbar \rho_0 c_p \sqrt{1 - V^2}$, in agreement with the findings of Ref. [16]. Note however that the existence of polarization solitons of the form (39) and (46) is not restricted to the condition of equal background densities $\rho_{\uparrow 0} = \rho_{\downarrow 0}$ considered in Ref. [16].

Our approach made it possible to identify new algebraic solitons (42) and (49) with unique properties which we now briefly discuss. The algebraic soliton (49) can be obtained as the limit $w_2 (= w_3) \to w_1$ of (46). It corresponds to an increased number of "up" particles $\Delta N_\uparrow = \pi \rho_0 \xi_p$. At variance with the case of dark/bright solitons, once the background parameters $w_0$ and $v_0$ are fixed, the velocity $V$ of an algebraic soliton is not free. One finds that it is fixed to be exactly the sound velocity (26). For an algebraic soliton, one has $w_2 \to w_1$ and thus the energy (54) of such a soliton is zero, as can be checked directly from (49) and (53).

Also note that the dark/bright solitons (39) and (46) are of a quite different nature than the one identified by Busch and Anglin in Ref. [26] and observed in Ref. [27]. It can be shown that if one considers the limit of a stationnary soliton of type (46) with no pedestal ($w_0 \to -1$), then one does not reach the limit of the dark-bright solitons of Ref. [26], but instead one obtains an algebraic soliton of the form $\rho_\uparrow(\zeta, \tau) = \rho_0 (1 + \zeta^2)^{-1}$.

# 4 Dispersionless approximation and simple-waves

## 4.1 Dispersionless hydrodynamics and Riemann equations

If the velocity and density distributions $v$ and $w$ are smooth enough, that is, if they experience little change over one polarization healing length (7), then we can neglect the dispersion effects described by the last terms in the second equations of the systems (21) and (23)[2] This corresponds to the so-called dispersionless approximation. We shall present the corresponding equations in two forms—for the variables $(w, v)$,

$$w_\tau - [(1 - w^2)v]_\zeta = 0, \quad v_\tau - [(1 - v^2)w]_\zeta = 0, \tag{57}$$

and for the variables $(\theta, \sigma)$,

$$\begin{aligned} \theta_\tau + 2\cos\theta \cdot \cos\sigma \cdot \theta_\zeta - \sin\theta \cdot \sin\sigma \cdot \sigma_\zeta = 0, \\ \sigma_\tau + 2\cos\theta \cdot \cos\sigma \cdot \sigma_\zeta - \sin\theta \cdot \sin\sigma \cdot \theta_\zeta = 0. \end{aligned} \tag{58}$$

These are equations of hydrodynamic type which can be studied by means of well documented methods.

First of all, we find at once from the system (58) that the variables

$$r_1 = \sigma - \theta, \quad \text{and} \quad r_2 = \sigma + \theta \tag{59}$$

satisfy the equations

$$\frac{\partial r_{1,2}}{\partial \tau} + V_{1,2}(r_1, r_2)\frac{\partial r_{1,2}}{\partial \zeta} = 0, \tag{60}$$

where

$$V_{1,2} = \frac{3}{2}\cos r_{1,2} + \frac{1}{2}\cos r_{2,1} = 2\cos\sigma\cos\theta \pm \sin\sigma\sin\theta, \tag{61}$$

or in terms of the variables $(v, w)$

$$V_{1,2} = 2wv \pm \sqrt{(1 - w^2)(1 - v^2)}. \tag{62}$$

---

[2]In this regime, the dispersion relation (24) can be approximated by a straight line of slope $c$ [$c$ being the speed of sound, as given by (26)], which is legitimate when $k \ll 1$, i.e., for wave lengths large compared to $\xi_p$.

The characteristic velocities $V_1$ and $V_2$ are the velocities of propagation of small disturbances along a non-uniform background $(\theta, \sigma)$ or $(w, v)$, correspondingly. In the case of a uniform background $w = w_0 = \cos\theta_0$, $v = v_0 = \cos\sigma_0$ they coincide with the sound velocities (26). The variables $r_{1,2}$ are called Riemann invariants, and Eqs. (60) are the hydrodynamic equations written in the Riemann invariant form (see, e.g., Ref. [28]). They have the familiar form of equations of compressible gas dynamics written in terms of the Riemann invariants, however the relationships between the Riemann invariants and the physical variables are more complicated here than for a gaseous system. Once $r_1$ and $r_2$ have been found, the physical variables $w$, $v$ are given by

$$w = \cos[(r_1 - r_2)/2], \quad v = \cos[(r_1 + r_2)/2]. \tag{63}$$

At this point, we have reduced the polarization hydrodynamic equations to the symmetric Riemann form (60). We shall now study a special class of solutions of these equations.

## 4.2 Simple wave solutions

In the framework of the hydrodynamic approximation a special role is played by the so-called *simple wave* solutions that are characterized by the fact that one of the Riemann invariants (59) is constant along the solution, so that the system (60) reduces to a single equation of the Hopf type. For example, let $r_2 = r_2^0 = \text{const}$; then we get the equation

$$\frac{\partial r_1}{\partial \tau} + V_1(r_1, r_2^0)\frac{\partial r_1}{\partial \zeta} = 0 \tag{64}$$

for the variable $r_1$. This equation admits the well-known solution

$$\zeta - V_1(r_1, r_2^0)\tau = f(r_1), \tag{65}$$

where $f(r_1)$ is an arbitrary function. Equation (65) determines the dependence of $r_1$ on $\zeta$ and $\tau$ in an implicit form. The function $f(r_1)$ can be thought of as the inverse function of the initial distribution of $r_1$ at the moment $\tau = 0$, i.e., $f^{-1}(\zeta) = r_1(\zeta, \tau = 0)$. The simple wave solution with constant Riemann invariant $r_1 = r_1^0 = \text{const}$ can be easily written in a similar form.

The importance of the simple wave solutions is related to the fact that, generally speaking, a hydrodynamic solution of a typical problem consists of different functions defined on several regions in the $(\zeta, \tau)$-plane separated by lines of discontinuity of the fields (here $\theta$ and $\sigma$). Along the so-called *weak discontinuities* one has discontinuities of the derivatives while the functions remain continuous. In particular, if the fluid flow has a boundary with adjacent quiescent fluid, then this boundary is a weak discontinuity and the neighboring flow is described by a simple wave solution (see, e.g., [28]).

A special role is played by self-similar solutions, for which $r_{1,2}$ depend on the self-similar variable $z = \zeta/\tau$ only. In particular, such solutions appear in problems where the initial distributions do not contain parameters with dimension of a length, e.g., in the case of initial discontinuities with abrupt jumps of the variables $w$ and/or $v$ ($\theta$ and/or $\sigma$). The jump occurs at some coordinate that can be taken as the origin of the $\zeta$-coordinate frame. In this case, $r_{1,2} = r_{1,2}(z)$ and the hydrodynamic equations (60) take the form

$$(V_1 - z)\frac{dr_1}{dz} = 0, \quad (V_2 - z)\frac{dr_2}{dz} = 0. \tag{66}$$

Their solutions are evidently

$$
\begin{aligned}
r_2 = r_2^0 = \text{const}, \quad &\text{and} \quad V_1(r_1, r_2^0) = z, \\
\text{or} \quad r_1 = r_1^0 = \text{const}, \quad &\text{and} \quad V_2(r_1^0, r_2) = z.
\end{aligned}
\tag{67}
$$

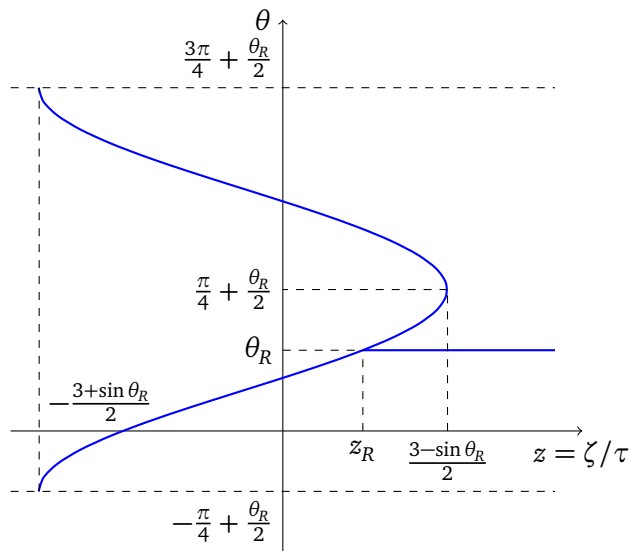

Figure 1: Distribution of $\theta(z)$ in the simple wave solution with fixed value of $r_2 = \sigma + \theta = \pi/2 + \theta_R$. The flow is attached on its right side to a condensate at rest with $\theta = \theta_R$ and $\sigma = \pi/2$ which corresponds to the horizontal line. Here $z_R = \sin\theta_R$.

These are particular cases of simple wave solutions (65) with $f \equiv 0$. Eqs. (67) yield for the variable $\theta$ the distributions

$$
\begin{aligned}
\theta &= \pm\frac{1}{2}\arccos\left(\frac{2}{3}z - \frac{1}{3}\cos r_2^0\right) + \frac{1}{2}r_2^0 + n\pi, \\
\text{or}\quad \theta &= \pm\frac{1}{2}\arccos\left(\frac{2}{3}z - \frac{1}{3}\cos r_1^0\right) - \frac{1}{2}r_1^0 + n\pi,
\end{aligned}
\tag{68}
$$

where the values of the constants ($r_2^0$ or $r_1^0$ and $n \in \mathbb{Z}$) and the signs are to be determined from the boundary conditions.

Let us consider here such solutions in the case where a dispersionless polarization flow is neighboring a condensate at rest. We shall first consider a self-similar simple wave matching at its right side a quiescent condensate (i.e., with $\sigma = \pi/2$) where $\theta = \theta_R = \text{const}$. It is easy to see from simple considerations [28] that its right edge, being a weak discontinuity, must propagate to the right with the sound velocity $c = \sin\theta_R$ [cf., (26)]; that is, this self-similar flow has to satisfy the boundary condition $\theta = \theta_R$ at $z = z_R = \sin\theta_R$. Simple inspection shows that this is achieved by the first of solutions (68) (where $r_2 = \sigma + \theta = \pi/2 + \theta_R = \text{const}$) with a lower sign and $n = 0$. Hence, owing to the relation $\arccos x = \pi/2 - \arcsin x$, we obtain

$$
\theta = \frac{1}{2}\arcsin\left(\frac{2}{3}z + \frac{1}{3}\sin\theta_R\right) + \frac{1}{2}\theta_R,
\tag{69}
$$

and, consequently, by virtue of constancy of $r_2 = r_2^0 = \frac{\pi}{2} + \theta_R$,

$$
\sigma = \frac{1}{2}\pi + \theta_R - \theta.
\tag{70}
$$

It is usually supposed that $\theta$ takes values in the interval $0 \le \theta \le \pi$, however any interval of same length is suitable for the description of the physical variable $w = \cos\theta$. We shall use here the equivalent interval

$$
-\frac{1}{4}\pi + \frac{1}{2}\theta_R \le \theta \le \frac{3}{4}\pi + \frac{1}{2}\theta_R
\tag{71}
$$

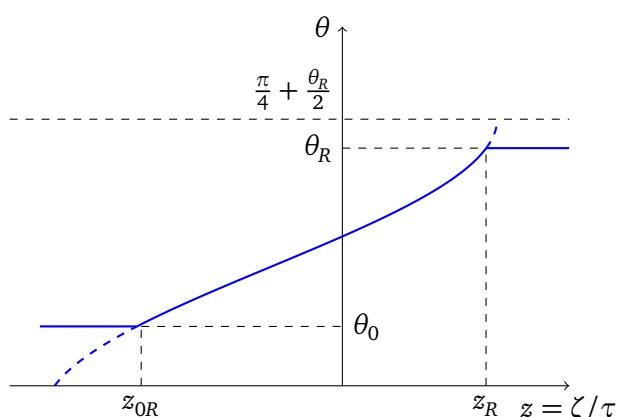

Figure 2: Distribution of $\theta(z)$ in the simple wave solution; case (a) (see (73)). Here $z_R = \sin\theta_R$, $z_{0R} = \frac{3}{2}\sin(2\theta_0 - \theta_R) - \frac{1}{2}\sin\theta_R$.

which is more suitable for the solution (69). The solution (69) does not cover all the interval (71) over which one has

$$z(\theta) = \frac{3}{2}\sin(2\theta - \theta_R) - \frac{1}{2}\sin\theta_R . \tag{72}$$

The resulting plot is displayed in Fig. 1 for a value of $\theta_R$ chosen in the interval $0 < \theta_R < \pi$.

The left edge of this wave must have a boundary either with one of the general solutions of equations (60), or with another simple wave with constant values of $\sigma$ and $\theta$ (that is, with a plateau in the density distribution). For future applications we shall confine ourselves to the second possibility and demand that the left edge of the solution corresponds to $\theta = \theta_0$ and, consequently, to $\sigma = \sigma_0 = \pi/2 + \theta_R - \theta_0$, since $r_2$ is constant across our simple wave. Here we have to distinguish two main typical situations denoted as (a) and (b) below.

Case (a): If

$$-\frac{1}{4}\pi + \frac{1}{2}\theta_R < \theta_0 < \theta_R, \tag{73}$$

then the constant left flow characterized by $\sigma_0$ and $\theta_0$ is connected with the quiescent condensate at the right by a rarefaction wave shown in Fig. 2 (region $z_{0R} < z < z_R$ of this figure) whose left edge propagates with velocity

$$z_{0R} = \frac{3}{2}\sin(2\theta_0 - \theta_R) - \frac{1}{2}\sin\theta_R. \tag{74}$$

The corresponding distributions of the density $\rho_\uparrow$ and the flow velocity $v = \cos\sigma = \sin(\theta - \theta_R)$ are shown in Figs. 3 and 4, respectively.

Case (b):

$$\theta_R < \theta_0 < \frac{3}{4}\pi + \frac{1}{2}\theta_R. \tag{75}$$

We will see that in this case there is an interval on the $z$-axis where the formal solution of the hydrodynamic equations becomes three-valued. Although such a solution does not have a direct physical meaning, it provides important relations remaining correct after replacement of the nonphysical multi-valued parts of the flow by a dispersive shock wave. To be definite, we illustrate such a situation in Fig. 5 which is drawn in the subcase we denote as (b1) in which

$$\theta_R < \theta_0 < \frac{1}{4}\pi + \frac{1}{2}\theta_R. \tag{76}$$

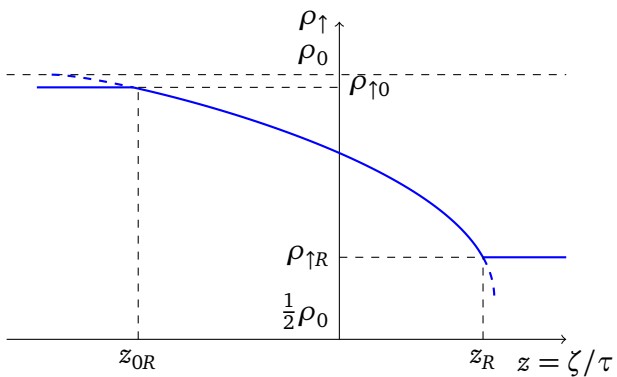

Figure 3: Distribution of $\rho_\uparrow(z)$ in the simple wave solution; case (a). Here $z_R = \sin\theta_R$, $z_{0R} = \frac{3}{2}\sin(2\theta_0 - \theta_R) - \frac{1}{2}\sin\theta_R$, $\rho_{\uparrow 0} = \rho_0\cos^2(\theta_0/2)$, $\rho_{\uparrow R} = \rho_0\cos^2(\theta_R/2)$.

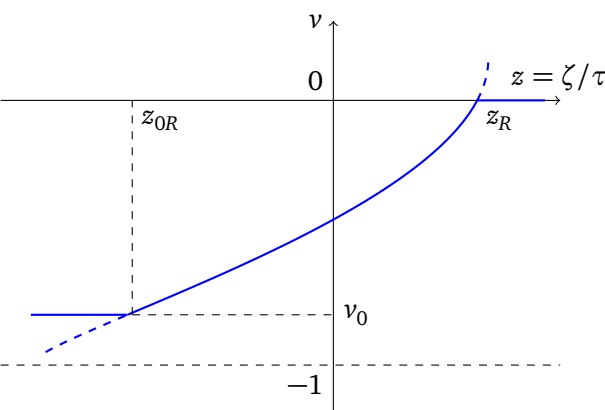

Figure 4: Distribution of $v(z)$ in the simple wave solution; case (a). Here $v_0 = \sin(\theta_0 - \theta_R)$.

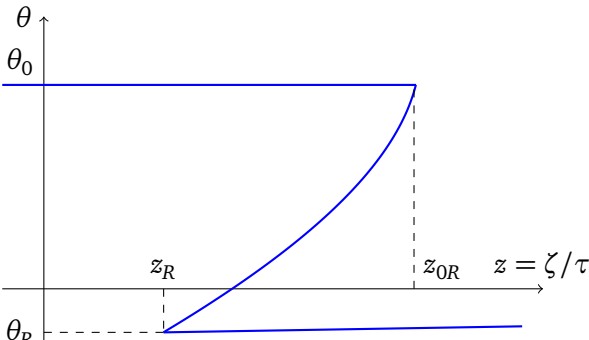

Figure 5: Distribution of $\theta(z)$ in the simple wave solution; case (b1) (see (76)). Here $z_R = \sin\theta_R$, $z_{0R} = \frac{3}{2}\sin(2\theta_0 - \theta_R) - \frac{1}{2}\sin\theta_R$.

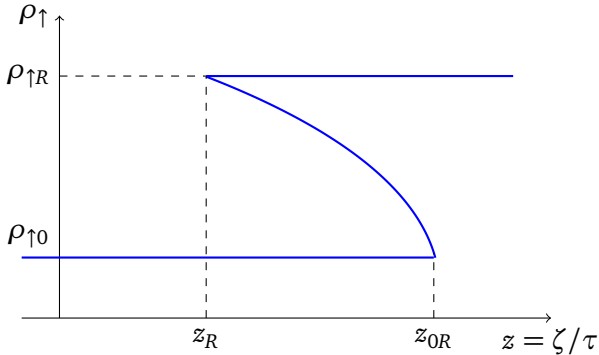

Figure 6: Distribution of $\rho_\uparrow(z)$ in the simple wave solution; case (b1). Here $z_R = \sin\theta_R$, $z_{0R} = \frac{3}{2}\sin(2\theta_0 - \theta_R) - \frac{1}{2}\sin\theta_R$.

In this case, in the region of the simple wave, $\theta(z)$ is given by the single-valued solution (69), but the matching with the left and right boundaries can only be performed at the price of overlapping the region of validity of the single wave solution with the ones of the plateau at the boundary. This corresponds to an overall multi-valued solution, as shown in Fig. 5. The corresponding plot of the density is shown in Fig. 6 and a similar graph can be plotted for the flow velocity $v(z)$.

In the subcase we denote as (b2) for which

$$\frac{1}{4}\pi + \frac{1}{2}\theta_R < \theta_0 < \frac{3}{4}\pi + \frac{1}{2}\theta_R, \tag{77}$$

the simple wave solution obtained from (72) already corresponds to a multi-valued $\theta(z)$ and the graphs of the formal hydrodynamic solutions can be easily plotted.

Let us now turn to a self-similar simple wave propagating to the left into a quiescent condensate with $\sigma = \pi/2$, $\theta = \theta_L = $ const. This problem is obviously symmetric to the one just studied: the left edge of the wave propagates here to the left with the sound velocity $c = -\sin\theta_L$ that is, we have to satisfy the boundary condition $\theta = \theta_L$ at $z = z_L = -\sin\theta_L$. This time we have to consider the second of solutions (68) (where $r_1 = \sigma - \theta = \pi/2 - \theta_L = $ const) with an upper sign and $n = 0$. Hence, we obtain

$$\theta = -\frac{1}{2}\arcsin\left(\frac{2}{3}z - \frac{1}{3}\sin\theta_L\right) + \frac{1}{2}\theta_L, \tag{78}$$

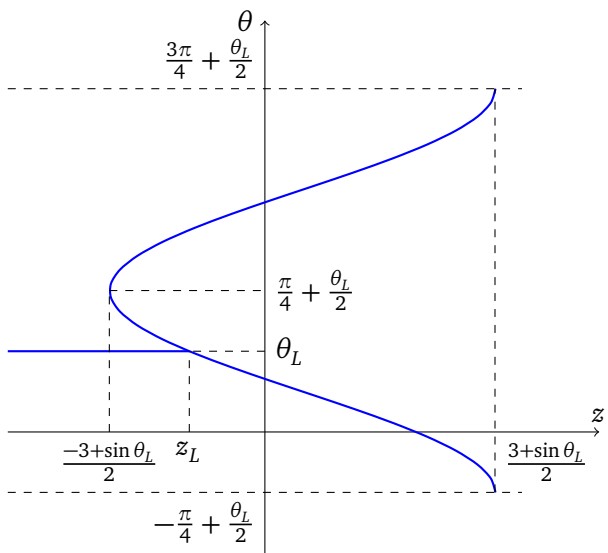

Figure 7: Distribution of $\theta(z)$ in the simple wave solution with fixed value of $r_1 = \sigma - \theta = \pi/2 - \theta_L$. The flow with $\theta = \theta_L$ and $\sigma = \pi/2$ (condensate at rest) can be attached to this solution on its left edge. It is shown by the horizontal line. Here $z_L = -\sin\theta_L$.

and, consequently,

$$\sigma = \frac{1}{2}\pi - \theta_L + \theta. \tag{79}$$

It is clear that the plots for this case can be obtained from the previous ones by the change $z \to -z$ replacing the notation $\theta_R \to \theta_L$, etc. Therefore we shall illustrate such a situation only by the plot of $\theta(z)$ which is displayed in Fig. 7.

Thus, we have obtained simple wave solutions which match on one boundary with a quiescent uniform condensate, and on the other with a flow with constant density and velocity—the "plateau solution".

Two important typical situations have been identified in this section. First, in some cases, the plateau solution can be connected to a simple wave solution joining a quiescent condensate on its other boundary. This is the situation illustrated in Figs. 3 and 4. For such flows the dispersionless hydrodynamic approach is indeed legitimate, and it is just expected that a more precise treatment of the weak discontinuities should exhibit a small amount of linear radiation (on both sides of the simple wave). Such flows are called *rarefaction waves*. Second, in some instances, the solution of the dispersionless hydrodynamic approach is multi-valued in some regions of space, cf. Fig. 6. In these regions, the physical flow is expected to be a *dispersive shock wave*, as commonly encountered in similar situations. In the next section we shall consider a configuration where these two possibilities are realized.

## 5 Evolution of a step-like discontinuity

As a typical application of the theory, let us consider an initial step-like distribution of polarization

$$\theta(\zeta, \tau = 0) = \begin{cases} \theta_L \,, & \text{when} \quad \zeta < 0 \,, \\ \theta_R \,, & \text{when} \quad \zeta > 0 \,. \end{cases} \tag{80}$$

and we assume here that the left and right asymptotic regions are both initially at rest,

$$\sigma(\zeta, \tau = 0) = \begin{cases} \sigma_L = \frac{\pi}{2}, & \text{when} \quad \zeta < 0, \\ \sigma_R = \frac{\pi}{2}, & \text{when} \quad \zeta > 0. \end{cases} \tag{81}$$

We shall consider this problem in the framework of the polarization dynamics governed by Eqs. (14), (21) or (23). We shall begin with the dispersionless hydrodynamic approximation corresponding to Eqs. (57) or (58) that can be written in the Riemann invariant form (60).

## 5.1 Hydrodynamic approximation

The step-like discontinuity evolves into a wave whose edges propagate into quiescent regions located at $\zeta \to \pm\infty$. If such an edge is represented by a weak discontinuity, then the adjacent flow is described by a simple wave solution. The step-like initial distribution (80) does not include any parameter having the dimension of a length and, consequently, the solution has to depend only on the self-similar variable $z = \zeta/\tau$ (and of course also, parametrically, on $\theta_L$ and $\theta_R$).

One cannot find a single simple wave joining its right and left boundaries with asymptotic regions corresponding to the initial conditions (80) and (81). Instead, the initial discontinuity evolves, for $\tau > 0$, into a more complex structure: an expanding self-similar wave consisting of two simple waves separated by a plateau characterized by the constant parameters $\theta_0$ and $\sigma_0$. One edge of each simple wave has a boundary with a condensates whose parameters are given by one (the left or the right) of the boundary conditions (80) and (81), the other edge matching the plateau distribution. As was discussed in the preceding section, along the simple wave solution [matching with the left asymptotic region $\sigma = \pi/2$, $\theta = \theta_L$] we have $r_1 = \sigma - \theta = \pi/2 - \theta_L = \sigma_0 - \theta_0$, and along the other simple wave solution [matching with the right asymptotic region $\sigma = \pi/2$, $\theta = \theta_R$] we have $r_2 = \sigma + \theta = \pi/2 + \theta_R = \sigma_0 + \theta_0$. These two conditions determine the parameters of the flow on the plateau:

$$\theta_0 = \frac{1}{2}(\theta_L + \theta_R), \quad \sigma_0 = \frac{1}{2}(\theta_R - \theta_L + \pi). \tag{82}$$

Combining with the simple wave solutions (whose characteristics are discussed in the previous section), we find the full solution of the problem – determined within the dispersionless approach – under the form

$$\theta(z) = \begin{cases} \theta_L, & z < z_L, \\ \frac{1}{2}\theta_L - \frac{1}{2}\arcsin\left(\frac{2}{3}z - \frac{1}{3}\sin\theta_L\right), & z \in (z_L, z_{0L}), \\ \frac{1}{2}(\theta_L + \theta_R), & z_{0L} < z < z_{0R}, \\ \frac{1}{2}\theta_R + \frac{1}{2}\arcsin\left(\frac{2}{3}z + \frac{1}{3}\sin\theta_R\right), & z \in (z_{0R}, z_R), \\ \theta_R, & z > z_R, \end{cases} \tag{83}$$

where

$$\begin{aligned} z_L &= -\sin\theta_L, \\ z_{0L} &= \frac{1}{2}\sin\theta_L - \frac{3}{2}\sin\theta_R, \\ z_{0R} &= \frac{3}{2}\sin\theta_L - \frac{1}{2}\sin\theta_R, \\ z_R &= \sin\theta_R. \end{aligned} \tag{84}$$

The edge at $\zeta = -z_L \cdot \tau$ propagates to the left at velocity $-\sin\theta_L$ which is the sound velocity in the left condensate. The edge at $\zeta = z_R \cdot \tau$ propagates to the right with velocity $\sin\theta_R$ which

is the sound velocity in the right condensate [cf. (26)], and the plateau is located between the edges $z_{0L} \cdot \tau \le \zeta \le z_{0R} \cdot \tau$.

Thus, for given values of the densities at both sides of the initial discontinuity (i.e. for given values of $\theta_L$ and $\theta_R$) one can calculate the parameters $\theta_0$, $\sigma_0$ defining the plateau distribution from (82) and determining the "left" and "right" simple wave solutions joining the quiescent condensates with the plateau. One of these simple waves represents a rarefaction wave and the other one describes a formal non-physical multi-valued solution. This means that the hydrodynamic approximation fails in the region where the flow is multi-valued and we have there to take into account the dispersion effects neglected in the long wavelength hydrodynamic theory. As a result of dispersion effects, the multi-valued region is replaced by a dispersive shock wave which is an oscillatory nonlinear wave structure. Such a situation is illustrated in Fig. 8. There the orange line describes the hydrodynamic approximation (83), for

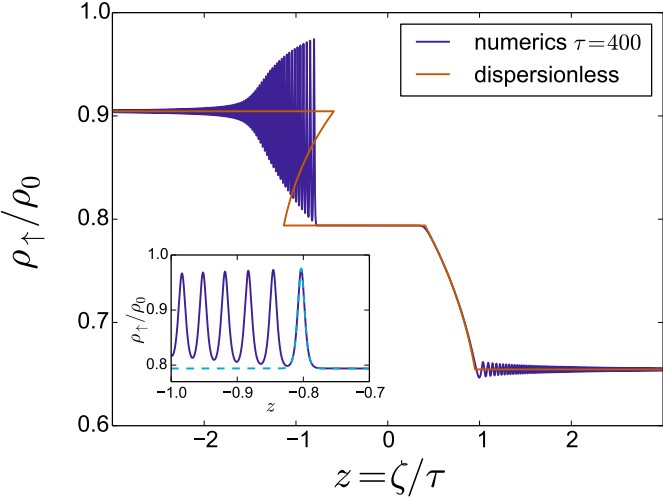

Figure 8: $\rho_\uparrow/\rho_0$ plotted as a function of $z = \zeta/\tau$ at $\tau = 400$. The initial profile is given by (85) and (86). The left and right asymptotic densities are $\rho_{\uparrow,L}/\rho_0 = \cos^2(\theta_L/2) = 0.9045$ and $\rho_{\uparrow,R}/\rho_0 = \cos^2(\theta_R/2) = 0.6545$. The dark blue curve corresponds to the numerical solution of Eqs. (14). The orange curve is the result of the dispersionless approximation. The inset displays the blow-up of the region of the soliton edge of the DSW. The dashed (light blue) line is the plot of the first soliton whose characteristics are determined in Sec. 5.2.

which the simple wave at the left of the plateau is multi-valued. The blue line corresponds to the numerical solution of the polarization dynamics equations (14) for an initial profile given by

$$v(\zeta, \tau = 0) = 0, \quad \text{and} \quad \theta(\zeta, \tau = 0) = \frac{\theta_R + \theta_L}{2} + \frac{\theta_R - \theta_L}{2} \tanh\left(\frac{\zeta}{\zeta_0}\right), \qquad (85)$$

with

$$\theta_L = 0.2\,\pi, \quad \theta_R = 0.4\,\pi, \quad \text{and} \quad \zeta_0 = 1. \qquad (86)$$

The value of $\zeta_0$ is not negligibly small, and the argument previously invoked for justifying the self-similar nature of the flow does not hold for all times. Instead, the structure of the flow – with a well defined plateau joined to both asymptotic regions by specific structures – does not appear instantaneously, but takes a finite amount of time to get formed. As a result, the flow can be considered as self-similar only for times large compared with this set-up time, which we numerically evaluate to be of order of $\tau_{\text{setup}} \simeq 8$ in the case of the initial conditions specified by (85) and (86).

It is clearly seen from Fig. 8 that both the right rarefaction wave and the plateau region are very well described by the hydrodynamic theory, the dispersion effects leading only to small oscillations in vicinity of the weak discontinuities located at the interface between these two regions. On the contrary, the region of large amplitude oscillations on the left side of the wave pattern is completely beyond reach of the dispersionless approach and in the next subsection we shall use a theory able to describe such dispersive shock wave (DSW) structures with account of dispersion effects.

## 5.2  Whitham modulation theory and Gurevich-Pitaevskii problem

As seen in Fig. 8, the numerical solution suggests that the dispersive shock wave can be seen as a nonlinear periodic solution of the polarization equations – such as those studied in Section 3 – which is however modulated, as shown by the fact that the amplitude of the oscillations is not constant. This modulation is gentle, in the sense that the parameters (amplitude, velocity, period, etc.) of the wave change little over one wavelength and one period of oscillation. This means that we can apply the Whitham averaging method for the description of this structure. In his original paper [29], Whitham assumed that the evolution of slowly modulated nonlinear waves can be described by equations obtained by averaging the densities and fluxes of the conservation laws over the rapid oscillations of the wave. He derived these averaged equations for several nonlinear wave equations, in particular, for the case of cnoidal wave solutions of the celebrated Korteweg-de Vries (KdV) equation, and—what was most remarkable from a mathematical point of view—he succeeded in transforming these equations into a diagonal Riemann form analogous to equations (60) obtained in the dispersionless approximation of hydrodynamic flows. As it became clear later, this success was related to the specific mathematical properties—complete integrability—of the KdV equation.

For the case we are interested in, a most important application of the Whitham theory was suggested by Gurevich and Pitaevskii [30]. In their approach it was assumed that the expanding DSW which develops after wave breaking can be described by the nonlinear periodic solution of the wave equation provided the parameters of this solution change slowly with time and space coordinate. They illustrated the method by applying it to the evolution of an initial step-like discontinuity and to the formation of a DSW after the wave breaking moment for the KdV wave dynamics.

Since the publications of the work of Whitham and Gurevich and Pitaevskii, the Whitham theory has been considerably developed in different directions and has found many applications in nonlinear physics. In particular, it was shown that many problems can be reduced to the consideration of the evolution of an initial step-like discontinuity. It was therefore of great importance to discover [31] that, for this specific step-like problem, the main characteristics of DSWs can be obtained by a simple method applicable to both completely integrable and non-integrable nonlinear wave equations. In our case the polarization wave dynamics is governed by the 1D version of the dissipationless Landau-Lifshitz equation which is completely integrable (see, e.g., [32]). However, the Whitham theory is not developed well enough for this equation and therefore El's method [31] seems the most appropriate for the description of the DSW observed in Fig. 8.

We thus assume that, instead of the multi-valued solutions found in the dispersionless approximation in the preceding subsection, a DSW is generated that joins the neighboring quiescent condensate at the left side of the wave structure with the plateau region. For definiteness, and in accordance with the example shown in Fig. 8, we consider the case where the Riemann invariant $r_1 = \sigma - \theta$ is constant across the multi-valued region. As was assumed by Gurevich and Meshcherkin [33] – and confirmed in many particular cases – one of the Riemann invariants preserves its value even after replacement of the multi-valued solution by the oscillatory DSW: in a sense, an equality of the type $r_1|_- = r_1|_+$ replaces in the case of DSWs

the well-known Rankine-Hugoniot relation of the theory of viscous shocks. It is then natural to assume that this relation is preserved by the Whitham averaging method, which yields an appropriate interpolation between the two edges of the DSW.

As we know, at the small-amplitude edge the DSW can be approximated by a modulated linear wave (47), however now propagating along a non-uniform background corresponding to the simple wave solution with $r_1 = \sigma - \theta = \pi/2 - \theta_L = \text{const}$, where we have used the values of the parameters at the left edge that matches with the left boundary conditions. With help of this relation we can write $\sigma = \pi/2 - (\theta_L - \theta)$ in the dispersion relation (25), leading to

$$\Omega(k, \theta) = -\left[2\sin(\theta - \theta_L) \cdot \cos\theta + \sqrt{\cos^2(\theta - \theta_L)\sin^2\theta + k^2}\right]k\,. \tag{87}$$

In (87) we have chosen the minus sign in front of the square root of (25) because we consider wave propagating to the left with respect to the background condensate. Equation (87) is the dispersion of linear waves propagating along a non-uniform $\theta$-distribution. During the smooth evolution of the oscillatory structure the local "number of waves" is preserved [34] which is expressed by the equation

$$k_\tau + \Omega_\zeta = 0\,. \tag{88}$$

Following El [31], we make a simple-wave type of assumption: in the DSW the wave number $k$ is a function of $\theta$ only, $k = k(\theta)$. Then, with account of (87), the law (88) of conservation of number of waves can be written under the form

$$\frac{dk}{d\theta} \cdot \theta_\tau + \left(\frac{\partial\Omega}{\partial k} \cdot \frac{dk}{d\theta} + \frac{\partial\Omega}{\partial\theta}\right)\theta_\zeta = 0. \tag{89}$$

On the other hand, substitution of $\sigma = \pi/2 + \theta - \theta_L$ into the first of equations (23) yields

$$\theta_\tau + \mathcal{V} \cdot \theta_\zeta = 0, \quad \text{where} \quad \mathcal{V} = -[2\sin(\theta - \theta_L)\cos\theta + \cos(\theta - \theta_L)\sin\theta]. \tag{90}$$

Imposing consistency of (89) and (90) considered as equations for $\theta$, we get

$$\frac{dk}{d\theta} = \frac{\partial\Omega/\partial\theta}{\mathcal{V} - \partial\Omega/\partial k}. \tag{91}$$

This is El's equation that can be extrapolated into the large amplitude nonlinear region by imposing the condition that the wavelength tends to infinity at the soliton edge, that is

$$k = 0 \quad \text{at} \quad \theta = \theta_0 = (\theta_L + \theta_R)/2. \tag{92}$$

Introducing the function

$$\alpha(\theta) = \sqrt{1 + \frac{k^2}{\cos^2(\theta - \theta_L)\sin^2\theta}}, \tag{93}$$

makes it possible to cast equation (91) into the form

$$\frac{d\alpha}{\alpha + 1} = \left(\frac{\sin(\theta - \theta_L)}{\cos(\theta - \theta_L)} - \frac{\cos\theta}{\sin\theta}\right)d\theta\,, \tag{94}$$

whose solution—with account of the boundary condition (92)—reads

$$\alpha(\theta) = \frac{\sin\theta_L + \sin\theta_R}{\cos(\theta - \theta_L)\sin\theta} - 1. \tag{95}$$

This yields

$$k(\theta_L) = \sqrt{\sin^2\theta_R - \sin^2\theta_L}\,. \tag{96}$$

Consequently, the left edge of the DSW propagates with the group velocity evaluated at $k(\theta_L)$:

$$v_{gr} = \left.\frac{\partial \Omega}{\partial k}\right|_{k(\theta_L)} = -\frac{2\sin^2\theta_R - \sin^2\theta_L}{\sin\theta_R}. \tag{97}$$

At the soliton edge of the DSW, we use the "soliton dispersion law" [31]

$$\widetilde{\Omega}(\kappa, \theta) = -\left[2\sin(\theta - \theta_L)\cdot\cos\theta + \sqrt{\cos^2(\theta-\theta_L)\sin^2\theta - \kappa^2}\right]\kappa \tag{98}$$

relating the velocity $V = \widetilde{\Omega}/\kappa$ of the soliton with the inverse width $\kappa$ that describes the exponential profile $w \cong w_3 + \frac{1}{2}(w_4 - w_3)\exp\{-\kappa|\zeta + V\tau|\}$ of the soliton far away from its center (in the regime $|\zeta| \to \infty$). The relation (98) follows from the remark that the soliton's tail propagates with the same velocity as the soliton itself and therefore the soliton's velocity can be found from the asymptotic behavior of its profile, see, e.g., [35,36]. Again following El, we assume that along the shock $\kappa = \kappa(\theta)$. Then the following equation can be derived (see [31]) for this function:

$$\frac{d\kappa}{d\theta} = \frac{\partial\widetilde{\Omega}/\partial\theta}{\mathcal{V} - \partial\widetilde{\Omega}/\partial\kappa}. \tag{99}$$

If we extrapolate the solution of (99) to the small amplitude region where $\kappa$ tends to zero, we obtain the boundary condition

$$\kappa(\theta_L) = 0. \tag{100}$$

Similarly to what has been done for the leading edge of the DSW [Eq. (91)], it is convenient for solving Eq. (99) to introduce the auxiliary function

$$\tilde{\alpha}(\theta) = \sqrt{1 - \frac{\kappa^2}{\cos^2(\theta - \theta_L)\sin^2\theta}}. \tag{101}$$

Inserting (101) into (99) and taking into account the boundary condition (100) one obtains

$$\tilde{\alpha}(\theta) = \frac{2\sin\theta_L}{\cos(\theta - \theta_L)\sin\theta} - 1. \tag{102}$$

Then, at the soliton edge, $\tilde{\alpha}$ is equal to

$$\tilde{\alpha}(\theta_0) = \frac{4\sin\theta_L}{\sin\theta_L + \sin\theta_R} - 1,$$

and, consequently, this edge propagates with velocity

$$V_s = \frac{\widetilde{\Omega}(\kappa(\theta_0), \theta_0)}{\kappa(\theta_0)} = -\frac{1}{2}(\sin\theta_L + \sin\theta_R). \tag{103}$$

The comparison of the analytic predictions (97) and (103) for the velocities of the edges of the dispersive shock wave with our numerical simulations is easily done for the well defined soliton edge, because, indeed, a leading soliton is easily identified at this edge of the numerically determined DSW. The velocity of this soliton tends for large time to the theoretical value, as illustrated in Fig. 9. In this figure, the numerical result for the velocity $V(\tau)$ of the soliton at the interface between the DSW and the plateau region is fitted with the empirical formula $V(\tau) = V_s^{\text{fit}} + b\,\tau^{-a}$, where $V_s^{\text{fit}}$, $a$ and $b$ are fitting parameters. At $\tau = 400$, $V$ is still off by about 5% from its asymptotic value, but the trend is in excellent agreement with the prediction (103) since one obtains $V_s^{\text{fit}} = -0.764$ whereas from (103) one expects $V_s^{\text{theo}} = -0.769$. The fitting procedure yields for the other parameters the values $a = 0.74$ and $b = -3.34$. Knowing

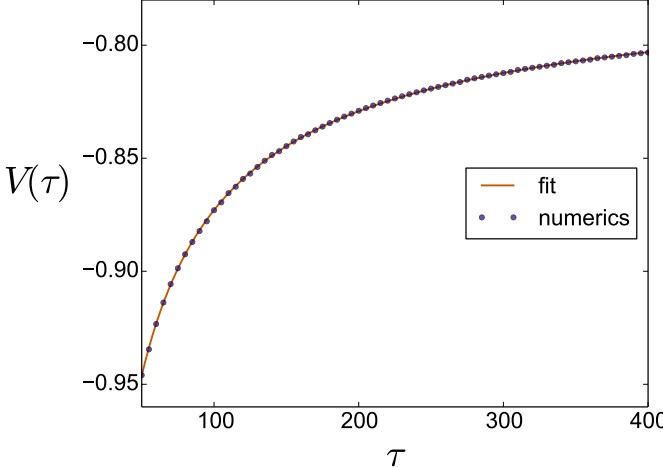

Figure 9: Dots: numerically determined velocity $V(\tau)$ of the trailing edge of the numerical solution. The initial conditions are specified in Eq. (85) and (86). Continuous line: fit of the numerical datas by the formula: $V(\tau) = V_s^{\text{fit}} + b\,\tau^{-a}$. One obtains $V_s^{\text{fit}} = -0.764$, in close agreement with the theoretical prediction from Eq. (103): $V_s^{\text{theo}} = -0.769$.

the velocity of the trailing edge soliton and the velocity and density of the background plateau over which it propagates, one can determine from (52) all the parameters $w_1$, $w_2 = w_3$ and $w_4$ characterizing the soliton. Again, the corresponding theoretical profile (46) is in excellent agreement with the numerics, as shown in the inset of Fig. 8. Note that whereas the shape and velocity of the soliton match the numerics, its position is not exactly the one expected for a purely self-similar flow (in which case it would be $z = V_s = -0.769$): this is to be related to the finite set-up time for creation the flow structure, cf. the discussion presented at the end of section 5.1 [after Eq. (86)].

As one can see in Fig. 8, it is difficult from the numerical solution to unambiguously locate the dispersive edge of the shock. Hence, at variance with the situation for the soliton edge, the velocity of the dispersive edge cannot be precisely extracted from the numerical simulation. However, one can reasonably argue that the value $v_{\text{gr}} = -1.54$ obtained from the theoretical formula (97) for the initial datas (86) matches quite well with the numerical results (cf. Fig. 8).

## 6  Discussion

In this section we discuss the accuracy of the polarization description of the dynamics of a two-component BEC [Eqs. (14)] and also the relevance of our approach to experimental studies.

A first question can be asked: in which extend does the assumption of decoupled dynamics apply? In other words, how small should $\delta g/g$ be in order for the approach followed in the present work to apply? A simple way for answering this question is to compare the results obtained from (14) with the ones obtained from the numerical solution of the full Gross-Pitaevskii system (1). This is done in Fig. 10 which displays the evolution of an initial profile of type (85). As one can see from this plot, the agreement is reasonable already for $\delta g/g = 0.2$ and becomes quite good for $\delta g/g = 0.05$. The lower part of the Figure shows that the assumption of constant total density is verified with an accuracy of order of 0.5% for $\delta g/g = 0.05$. We note that the largest departure of the total density from a constant occurs when $\rho_\uparrow/\rho_0$ is close to unity, i.e., when $\theta$ is close to 0, as anticipated in Eq. (11). Note also that the spatial and time scales ($\xi_p$ and $T_p$) are quite relevant: the Gross-Pitaevskii system is solved for

quite different values of these characteristic scales (the value of $\xi_p$ is multiplied by a factor 2 and the one of $T_p$ by a factor 4 when one goes from $\delta g/g = 0.2$ to $\delta g/g = 0.05$), but after the same time expressed in units of $T_p$ (24 $T_p$ in the case of Fig. 10), the spatial structures almost overlap if the appropriate units are used.

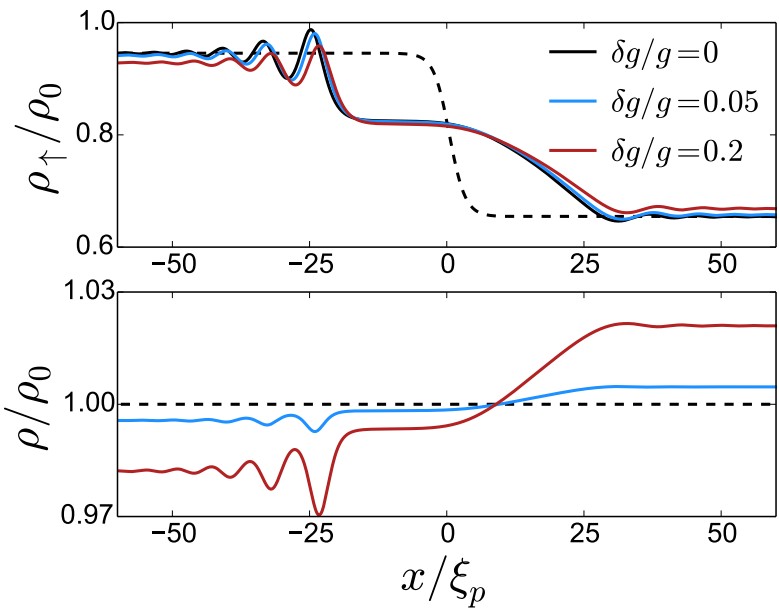

Figure 10: Upper plot: The black solid line represents $\rho_\uparrow$ as a function of position as obtained from solving the system (14) for the initial condition (85) with $\theta_L = 0.15\pi$, $\theta_R = 0.4\pi$ and $\zeta_0 = 3$ (dashed line). The numerical solution of the Gross-Pitaevskii system for the same initial condition and different values of $\delta g/g$ is represented by colored lines. Lower plot: same as above for the total density $\rho$.

Another question naturally arises: since Bose-Einstein condensation of ultra-cold atomic vapors is always realized in trapped systems, it is important to evaluate the experimental relevance of the infinitely extended configuration studied in the present work. One can first state that the theory has a physical meaning as long as its characteristic length $\xi_p$ (7) is much less that the size $X$ of spatial overlap of the two components which can be estimated in the framework of the Thomas-Fermi approximation presented in Appendix B :

$$\xi_p \ll X = \frac{\sqrt{g\rho_0}}{\omega_\| \sqrt{m}}, \qquad (104)$$

where $\omega_\|$ is the longitudinal trapping angular frequency and $\rho_0 \simeq N/X$, $N$ being the total number of atoms. The condition (104) combined with (2) reads

$$\frac{m\omega_\|^2 \xi^2}{\rho_0 g} \ll \frac{\delta g}{g} \ll 1, \qquad (105)$$

where $\xi = \hbar/\sqrt{2m\rho_0 g}$ is the healing length ($\xi_p = \xi\sqrt{g/\delta g}$). The first inequality of (105) can be also rewritten as

$$\omega_\| \xi \ll c_p \quad \text{or} \quad \frac{\xi}{c_p} \ll \frac{1}{\omega_\|}, \qquad (106)$$

that is the polarization sound velocity must be much greater than the healing length divided by the period of oscillations of atoms in the trap, or, in other words, the polarization wave passes the healing length in a time much less that the period of oscillations in the trap.

It is also worthwhile to address another point: it is known [37–39] that, in the presence of a trapping potential, the condition of uniform miscibility (which, in our notations, reads $\delta g > 0$) is not sufficient to ensure a good spatial overlap of the two components. This point is discussed in Appendix B where it is shown that, close to the mixing-demixing transition, the trapping potential induces a kind of phase separation if the lower of the intra-species nonlinear constants (say $g_{\downarrow\downarrow}$) is smaller than the inter-species constant $g_{\uparrow\downarrow}$, although the criterion of uniform miscibility $g_{\uparrow\downarrow} < \sqrt{g_{\uparrow\uparrow} g_{\downarrow\downarrow}}$ is (weakly) fulfilled.

This phenomenon could explain why, in Ref. [40], a kind of phase separation is observed in the mixture of the two hyperfine states $|\downarrow\rangle = |F = 1, m_F = -1\rangle$ and $|\uparrow\rangle = |F = 1, m_F = 0\rangle$ of $^{87}$Rb in spite of fulfilment of the uniform mixing condition. For this system $(a_{\uparrow\uparrow}, a_{\downarrow\downarrow}, a_{\uparrow\downarrow}) = (100.86\, a_0, 100.4\, a_0, 100.41\, a_0)$, where $a_0$ is the Bohr radius. Thus $a_{\downarrow\downarrow} < a_{\uparrow\downarrow}$ which implies mixing of the components in a uniform case; but non-uniformity caused by the trap potential induces phase separation. Instead, for the mixture of the two hyperfine states $|\uparrow\rangle = |F = 1, m_F = -1\rangle$ and $|\downarrow\rangle = |F = 2, m_F = -2\rangle$ of $^{87}$Rb one has $(a_{\uparrow\uparrow}, a_{\downarrow\downarrow}, a_{\uparrow\downarrow}) = (100.4\, a_0, 98.98\, a_0, 98.98\, a_0)$, that is the criterion on miscibility is also fulfilled, but here $a_{\downarrow\downarrow} = a_{\uparrow\downarrow}$ and the authors observe a large region of overlap of the two components.

Finally, concerning the comparison of our results with the ones presented in Ref. [20], it is worth noticing that if $\theta_L \to 0$, that is $\rho_{\uparrow L} \to 1$, then the left edge group velocity (97) tends to the value $v_{gr} = -2\sin\theta_R$ which coincides with the limiting value of velocity (74) of the left edge of the rarefaction wave $z_{0R}$ corresponding to $\theta_0 = 0$. This means that the DSW pattern is represented by small amplitude oscillations around the extrapolation of the rarefaction wave to the region with $\theta_L \to 0$, $\rho_{\uparrow L} \to 1$. As a result, the pattern looks like the rarefaction wave connecting two regions of quiescent condensates with different values of $\theta$: $\theta_L = 0$ and $\theta_R \neq 0$. This apparently agrees with the numerical simulations of the so-called subcritical regime discussed in [20] where only the rarefaction wave was observed for small enough values of the relative velocity and $\rho_{\uparrow L} = 1$.

# 7 Conclusion

In vicinity of the mixing/demixing transition, in the limit (2) first identified in Ref. [16], the polarization dynamics decouples from density waves and is described by the universal equations (14). In this paper we have identified new specific polarization structures associated with these equations in the case of a one dimensional system: algebraic solitons, simple waves, dispersive shock waves, etc. But more remains to be done. For instance, the non-monotonous behavior of the Riemann velocities (cf. section 4.2) is typically associated to a rich variety of different types of shocks [24] which remain to be investigated in the case at hand; in particular for situations with large jumps of the parameter $\theta$, when DSWs consisting of combined cnoidal and trigonometric parts are expected. The precise behavior of algebraic solitons in several instances, and a reliable procedure for their physical implementation would also be of great interest. The configuration described by the initial distributions (80) and (81) is too schematic for being able to describe the experiments presented in [20] where regions with different density ratios are colliding with finite initial relative velocities. One should thus consider the case where $\sigma_L$ and $\sigma_R$ are not both equal to $\pi/2$, and where the plateau formed after the collision is modulationnally unstable. Finally, the approach developed in this paper can be generalized to include Rabi coupling between the components (see, e.g., [41]) and also to two- or three-dimensional situations [42]. In particular, formation of oblique polarization solitons by the flow of the binary condensate past a polarized obstacle (see, e.g., [43]) can be considered in the framework of the present method. Works in these directions are in progress.

## Acknowledgements

We thank S. Stringari for fruitful discussions. AMK thanks Laboratoire de Physique Théorique et Modèles Statistiques (Université Paris-Sud, Orsay) where this work was started, for kind hospitality.

**Funding information** This work was supported by the French ANR under grant n° ANR-15-CE30-0017 (Haralab project).

## A Computation of the energy of a soliton

We briefly present here the computation leading to the result (54) for the energy of the soliton. From (27) and (28) one gets $\phi_\zeta = V(B-w)/(1-w^2)$ with $B = (1-w_2^2)v_0/V + w_2$ and from (30) $\theta_\zeta^2 = -Q(w)/(1-w^2)$. This yields for the energy (53)

$$\mathcal{E} = \int_{\mathbb{R}} \frac{d\zeta}{2} \left\{ \frac{-Q(w)}{1-w^2} + (1-w^2)\left[ V^2 \frac{(B-w)^2}{(1-w^2)^2} - 1 \right] + (1-v_0^2)(1-w_2^2) \right\} . \tag{107}$$

The integrand being symmetric —since $w(\zeta)$ is— one can thus restrict the range of integration to the domain $(-\infty, 0]$ over which one can write $d\zeta = +dw/\sqrt{-Q(w)}$. Using the fact that one can express $B$, $v_0$ and $V$ as functions of $w_1$, $w_2$ and $w_4$ [cf. Eq. (52)], it is then possible to re-write (107) under the form

$$\mathcal{E} = \int_{w_2}^{w_4} dw \frac{2w - w_2 - w_4}{\sqrt{(w_4 - w)(w - w_1)}} , \tag{108}$$

which yields the result (54).

## B Effective demixing in a 1D trap

In this appendix we present 1D computations in the framework of the Thomas-Fermi description of the system (1) in the presence of a trapping potential [44]. It is known [45] that the Thomas-Fermi approximation cannot quantitatively describe all the possible configurations encountered the mixture of two BECs, but it will permit to identify specific situations which will then have to be confirmed by a full numerical solution.

We consider here $N_\uparrow$ and $N_\downarrow$ atoms of each component placed in a harmonic potential of longitudinal angular frequency $\omega_\parallel$ much smaller than the radial trapping angular frequency $\omega_\perp$. In the so called "1D mean field regime" [46], the system can be described by the effective 1D Gross-Pitaevskii equation (1) with $g_{\uparrow\uparrow} = 2\hbar\omega_\perp a_{\uparrow\uparrow}$ [47] where $a_{\uparrow\uparrow}$ is the 3D intra-species $s$-wave scattering length of the "up" component (an similar expressions for $g_{\downarrow\downarrow}$ and $g_{\uparrow\downarrow}$). In the situation we are interested in where $N_\uparrow \sim N_\downarrow$ and $a_{\uparrow\uparrow} \sim a_{\uparrow\downarrow} \sim a_{\downarrow\downarrow}$, the 1D mean field regime holds when $N_\uparrow(\omega_\parallel/\omega_\perp)(a_{\uparrow\uparrow}/a_\perp) \ll 1$, where $a_\perp = \sqrt{\hbar/m\omega_\perp}$ is the radial harmonic oscillator length.

We chose the parameters so that the mean field condition of miscibility $\sqrt{a_{\uparrow\uparrow}a_{\downarrow\downarrow}} > a_{\uparrow\downarrow} > 0$ is always fulfilled, and in the following we denote as $A$ the parameter having the dimension of length defined by

$$A^2 = a_{\uparrow\uparrow}a_{\downarrow\downarrow} - a_{\uparrow\downarrow}^2 > 0 . \tag{109}$$

We define the non-dimensional position $X = x/a_\parallel$, where $a_\parallel = \sqrt{\hbar/m\omega_\parallel}$ is the longitudinal harmonic oscillator length, and the non-dimensional densities $n_{\uparrow,\downarrow}$ such that $\int n_{\uparrow,\downarrow}(X)dX =$

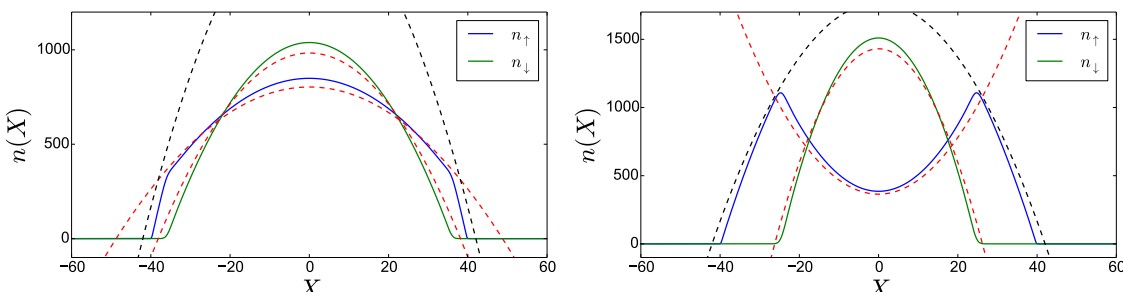

Figure 11: Distribution of atoms in a 1D trapped two-component BEC. The trap parameters are $\omega_\parallel = 2\pi \times 1\,\text{Hz}$, $\omega_\perp = 2\pi \times 500\,\text{Hz}$ and $N_\uparrow = N_\downarrow = 5 \times 10^4$. The condensates are formed by $^{87}\text{Rb}$ atoms, this yields $a_\parallel = 10.8\,\mu\text{m}$. The red dashed lines correspond to $n_\uparrow^a(x)$ and $n_\downarrow(x)$, Eqs. (113) and (111). The black dashed lines display $n_\uparrow^b(x)$ (114). The solid lines correspond to the numerical solution of the Gross-Pitaevskii equations. The left plot corresponds to the values $(a_{\uparrow\uparrow}, a_{\downarrow\downarrow}, a_{\uparrow\downarrow}) = (102\,a_0, 101\,a_0, 100\,a_0)$ for the scattering lengths. The right plot corresponds to $(a_{\uparrow\uparrow}, a_{\downarrow\downarrow}, a_{\uparrow\downarrow}) = (102\,a_0, 99\,a_0, 100\,a_0)$. The precise values of these scattering lengths have been chosen for exemplifying the phenomenon of effective demixing, but they all lie within a realistic range for $^{87}\text{Rb}$.

$N_{\uparrow,\downarrow}$. We denote as "down" the component for which the intra-species interaction is the lowest, i.e., $a_{\downarrow\downarrow} < a_{\uparrow\uparrow}$. Within the Thomas-Fermi approach one obtains

$$n_\uparrow(X) = \begin{cases} n_\uparrow^a(X) & \text{if } |X| \leq X_\downarrow, \\ n_\uparrow^b(X) & \text{if } X_\downarrow \leq |X| \leq X_\uparrow, \\ 0 & \text{if } X_\uparrow \leq |X|, \end{cases} \tag{110}$$

and

$$n_\downarrow(X) = \begin{cases} \frac{\omega_\parallel}{\omega_\perp} \frac{(a_{\uparrow\uparrow} - a_{\uparrow\downarrow})a_\parallel}{4A^2} \left(X_\downarrow^2 - X^2\right) & \text{if } |X| \leq X_\downarrow, \\ 0 & \text{if } X_\downarrow \leq |X|, \end{cases} \tag{111}$$

where

$$X_\uparrow^3 = \frac{3\,\omega_\perp}{\omega_\parallel} \frac{a_{\uparrow\uparrow}N_\uparrow + a_{\uparrow\downarrow}N_\downarrow}{a_\parallel}, \quad X_\downarrow^3 = \frac{3\,\omega_\perp}{\omega_\parallel} \frac{A^2 N_\downarrow}{(a_{\uparrow\uparrow} - a_{\uparrow\downarrow})a_\parallel}, \tag{112}$$

$$n_\uparrow^a(X) = \frac{\omega_\parallel}{\omega_\perp} \left[ \frac{a_\parallel}{4a_{\uparrow\uparrow}} X_\uparrow^2 - \frac{a_{\uparrow\downarrow}a_\parallel}{4A^2} \left(1 - \frac{a_{\uparrow\downarrow}}{a_{\uparrow\uparrow}}\right) X_\downarrow^2 - \frac{(a_{\downarrow\downarrow} - a_{\uparrow\downarrow})a_\parallel}{4A^2} X^2 \right], \tag{113}$$

and

$$n_\uparrow^b(X) = \frac{\omega_\parallel}{\omega_\perp} \frac{a_\parallel}{4a_{\uparrow\uparrow}} \left(X_\uparrow^2 - X^2\right). \tag{114}$$

These results are compared in Fig. 11 with the numerical solutions of Eqs. (1) in the presence of a trapping potential $V(x) = \frac{1}{2}m\omega_\parallel^2 x^2$. The two plots of this figure are drawn for a configuration verifying the miscibility condition (109) [3]. In the left plot $a_{\downarrow\downarrow} > a_{\uparrow\downarrow}$ whereas the situation is reversed in the right one (similar plots have already been obtained in Ref. [37]). Although the corresponding change of scattering lengths is minute, close to the mixing-demixing transition the effect is spectacular: one reaches a situation of quasi-demixing where the component with the largest scattering length (the up component) is expelled from the trap's center. This

---

[3]For the chosen sets of parameters, one is at the limit of the 1D mean field regime : $N_\uparrow(\omega_\parallel/\omega_\perp)(a_{\uparrow\uparrow}/a_\perp) \simeq 1$. The condition of applicability of the Thomas-Fermi approximation [46] is well fulfilled: $[N_\uparrow(a_{\uparrow\uparrow}/a_\perp)\sqrt{\omega_\perp/\omega_\parallel}]^{1/3} \simeq 23 \gg 1$.

situation would be expected in the situation $a_{\downarrow\downarrow} \ll a_{\downarrow\uparrow} \simeq a_{\uparrow\uparrow}$. The point is here that the same effect is observed for a system verifying the miscibility condition (109) provided one remains close to immiscibility and that $a_{\downarrow\downarrow} \lesssim a_{\downarrow\uparrow}$. The parameter governing the expulsion of the up component from the center of the trap is the non-dimensional curvature of its density at $X = 0$. From (113) this parameter is equal to

$$-\frac{\omega_\parallel}{\omega_\perp} \times \frac{(a_{\downarrow\downarrow} - a_{\uparrow\downarrow})a_\parallel}{a_{\uparrow\uparrow}a_{\downarrow\downarrow} - a_{\uparrow\downarrow}^2} \, . \tag{115}$$

In the cases presented in Fig. 11 the value of this parameter changes from $-1.3$ (in the left plot of the figure) to $+4.2$ (right plot) just by changing $a_{\downarrow\downarrow}$ by 2%.

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
