# Peer review of "Dispersive hydrodynamics of nonlinear polarization waves in two-component Bose-Einstein condensates"

_SciPost Physics, doi:SciPost Phys. 1, 006 (2016)_

## Round 1 · Referee Report · Anonymous (Referee 2) · 2016-9-12

Strengths

This is a very solid and systematic study of a particular subject, the polarization waves in a two-component Bose-condensate.

Weaknesses

The degree of novelty is reduced by the fact that this article builds heavily on the results obtained
in Ref. [15], albeit leading to empirically important generalizations, to the cases of unequal density and
unequal velocity of the components.

Report

Overall, this is a very good paper that deserves publishing. It sets a firm framework for further developments. The variety of the soliton solutions obtained indicates that a few interesting discoveries
are still awaiting ahead. The analysis in the dispersionless limit and the subsequent DSW treatment are
as novel as it is powerful, applications including. The Riemann invariants appear very naturally,
and I believe this is not the last instance in the two-component BEC that they do.

Requested changes

(*) In Eq. (3), the chemical potentials $\mu$ are undefined. I am sure those are the equilibrium chemical potentials without the inter-specie interaction, but it is not obvious;

(*) Right after Eq. 14: in "This shows that for time scales of order Tp and length scales of order ξp the polarization degree of freedom decouples from the density degree of freedom, even in the nonlinear regime." it is not clear what "[t]his" refers to. My understanding is that what is meant is that if one averages of the fast density oscillations, both in time and in space, h/she will be left with the purely polarization dynamics. But this was clear even before the Eq. 14;

(*) It would be desirable if authors can rigorously identify a dimensionless small parameter (or at least a dimensionful one) that generates an expansion truncation of which leads to (57) and (58). I do suspect that this is ratio between the polarization healing length and the size of a polarization feature, but I am not sure;

(*) The first time the Riemann invariants appear, a reference is in order, e.g. [L.D. Landau & E.M. Lifshitz, Fluid Mechanics (Pergamon Press (1959)), paragraph 104];

.............

(*) In "Conclusion", "DSW" is misspelled;

(*) There are many instances of missing commas, a native speaker may spot those instantly. The ones I noticed are:

  • After (3): "In this expression, " ;
  • After (14): "This shows that for time scales of order Tp and length scales of order ξp, the polarization degree of freedom decouples from the density degree of freedom, even in the nonlinear regime." ;
  • between (62) and (63): "The variables r1,2 are called Riemann invariants, and Eqs. (60) are the hydrodynamic equations written in the Riemann invariant form."

, but there are a few more.

  • validity: top
  • significance: high
  • originality: high
  • clarity: high
  • formatting: perfect
  • grammar: good

Author:  Thibault Congy  on 2016-10-10  [id 61]

(in reply to Report 2 on 2016-09-12)
Category:
answer to question

We thank the referee for his/her positive evaluation of our work and for relevant comments. Indeed the referee is right, we intend to pursue the study of this peculiar configuration of two-components BECs, when the intraspecies scattering lengths are slightly larger than the interspecies ones.

We now list the changes asked by the referee:

(*) we will explicitly give the values of the chemical potentials (this was also a request of the first referee).

(*) sentence after Eq. (14). Indeed in the special configuration we consider, the density and the polarization degrees of freedom correspond to very different length and time scales, and simply decouple one from the other. Averaging could be a way to kill density oscillations; time and length filtering could be an other one. A third way is, as considered in section 5, to implement situations where the density degree of freedom is initially not excited and, consequently, practically will not be excited during evolution of the polarization
wave. Indeed, if we excite a polarization wave of amplitude a, then, at it follows from Eq. (10), its evolution will lead to excitation of a density wave with amplitude \sim a(\delta g/g) << a which is negligibly small in our approximation, resulting in Eqs. (14) for the polarization dynamics. The effectiveness of the decoupling ensures that density waves will then barely be excited, as verified in Fig. 10. As demonstrated in Ref. [19] of the paper, the decoupling is quite efficient, even in the presence of an external potential which induces a position dependent density.

(*) The referee asks if there is a dimensionless parameter governing the validity of the dispersionless approximation. He/she then correctly identifies this parameter as being the ratio of the polarization healing length and the size of the polarization feature. The simpler way to convince oneself that this is indeed the
correct parameter is by checking (i) when do Eqs. (23) can be reduced to the dispersionless form (58) or also, (ii) when [i.e. for which wave vector] can the dispersion relation (24) be approximated by a straight line of slope c [c is the speed of sound, as given by (26)]. This last approximation is valid when k<<1, i.e. in the limit identified by the referee. This point was already rapidly mentioned in the beginning of section 4.1, and we add a footnote for clarity.

we thank the referee for all his/her other constructive remarks (in particular concerning the grammar), which do not need specific answers and which have been implemented in a revised version of the manuscript, to be soon uploaded on arXiv.

---

## Round 1 · Referee Report · Anonymous (Referee 1) · 2016-9-12

Strengths

  1. Original and relevant research

Weaknesses

  1. Definition and importance of algebraic solitons

Report

This is a very nice manuscript reporting new results on dynamics of spinor condensates. The authors consider polarisation dynamics in a system consisting of two species of bosons conveniently described by spin 1/2 and solve the corresponding coupled Gross-Pitaevskii equations in the limit where the inter-atomic coupling is much weaker than the coupling withing each component. Despite the system being one-dimensional which precludes existence of condensate in the usual thermodynamic sense, the physics considered here happens on the length scales of the order of healing length so that the classical treatment is valid.

Overall I recommend the manuscript for publication. The only point where I have doubts is the definition of the algebraic solitons. They correspond to the short length limit of the periodic waves described in section 3. I wonder whether it is an artefact of the limiting procedure which makes the structure look like a solitary wave. I wish it was explained better.

In addition there are some minor potential improvements:

  1. The sign of interactions corresponds to repulsion and this must be stated explicitly in the text.

  2. In Section 2 the chemical potential \mu_\up, \mu_\down are introduced and they seem to be related to the total density \rho_o, however the explicit relation is not stated and the difference between \rho and \rho_0 can only be guessed.

  3. Eq. (44) is identical to Eq. (35). I guess w_1 should be replaced by w_3.

Requested changes

  1. The sign of interactions corresponds to repulsion and this must be stated explicitly in the text.

  2. In Section 2 the chemical potential \mu_\up, \mu_\down are introduced and they seem to be related to the total density \rho_o, however the explicit relation is not stated and the difference between \rho and \rho_0 can only be guessed.

  3. Eq. (44) is identical to Eq. (35). I guess w_1 should be replaced by w_3.

  • validity: top
  • significance: high
  • originality: high
  • clarity: good
  • formatting: excellent
  • grammar: excellent

Author:  Thibault Congy  on 2016-10-10  [id 60]

(in reply to Report 1 on 2016-09-12)
Category:
answer to question

We thank the referee for his/her positive evaluation of our work and for careful reading of the manuscript. The referee raises a point concerning the algebraic solitons we have identified. He/she remarks that we have presented these solitons as limiting cases of trigonometric nonlinear waves: Eq. (41) leads to the algebraic soliton (42) [and Eq. (48) to (49)]. The referee then legitimately asks if the algebraic solitons are not spurious limits of a periodic structure. The answer is no. This can be seen by two different manners:

(a) the first and more straightforward one consists in verifying that the algebraic expression (42) is indeed a solution of the equations of motion (21). We made this check and indeed verified that (42) is a bona fide solution of (21).

(b) a second manner is already briefly mentioned in the manuscript, at the end of section 3. The algebraic soliton can also be obtained as a limiting case of the (usual) cosh^2 soliton (46) and thus indeed corresponds to a localized propagation moving a constant speed without deformation.

The referee then suggests 3 improvements of the text, which we will implement (together with a discussion of the above point concerning algebraic solitons) in an iterated version of the manuscript. We thank the referee for these relevant suggestions. Please note that Eqs. (44) and (35) do not exactly coincide: they are written in order that the square root at the denominator in positive.

---

## Round 1 · Referee Report · Anonymous (Referee 3) · 2016-9-19

Strengths

1) Timely: there are a number of on-going research projects in related directions 2) Well-written paper 3) The simpler problem of dispersive hydrodynamics of single component BECs has been extensively studied; this paper breaks open the study of two-component BEC dispersive hydrodynamics 4) Rich nonlinear physics: solitons, periodic traveling waves, rarefaction waves, and dispersive shock waves

Weaknesses

1) Some minor grammatical issues (see requested changes) 2) Could write a longer paper delving deeper into the nonlinear wave structures; this isn't really a weakness but rather a motivation for more study

Report

This timely paper considers the nonlinear polarization wave dynamics
in two-component BECs when the intraspecies scattering lengths are
slightly larger than the interspecies scattering length. This
assumption enables an important approximation of the governing vector
Gross-Pitaevskii equations as the dissipationless Landau-Lifshitz or
torque equation of magnetization dynamics. All the periodic and
soliton traveling wave solutions are explicitly characterized. The
long wavelength, dispersionless approximation of these equations is
also solved for rarefaction wave solutions in order to characterize
smooth counterflows. Finally, dispersive shock waves that occur when
the flow experiences wavebreaking are studied using Whitham nonlinear
wave modulation theory for the specific example of a sharp relative
density perturbation to quiescent condensates. Numerical simulations
validate the analytical results within their regimes of validity.

This well-written paper provides a description of an important line of
research on dispersive hydrodynamics of coupled BECs where dispersion
and nonlinearity lead to coherent structures, the most noteworthy of
which are solitons, rarefactions, and dispersive shock waves. The
dispersive hydrodynamics of single component BECs have received
significant attention so the corresponding study of the richer, more
complex case of two-component BECs is both natural and important.
While solitons for restricted set of parameter values have previously
been studied for this system, this paper provides their complete
characterization by direct integration of the governing equations.
Moreover, it also identifies nonlinear periodic traveling waves,
spatially extended dispersive shock waves, and rarefactions. In my
opinion, this is an important work that should be published with
haste.

Requested changes

Minor comments that the authors are free to consider or not:

1) According to Fig. 8, the small amplitude edge of the DSW is leftmost therefore the first sentence on p 22 should read "Consequently, the left edge of the DSW ..."

2) The authors obtain a prediction for the DSW soliton edge speed and compare with numerics. Although it is equivalent in some sense, they could additionally report and compare the soliton amplitude, which is determined by the soliton amplitude-speed relation.

3) It could be beneficial to the reader to mention the recent review article on dispersive shock waves [El, Hoefer, Physica D 2016].

4) The dispersionless equations (60) have the property that whenever r1 = r2, V1 = V2, i.e., the dispersionless system is nonstrictly hyperbolic at degenerate points (when one component has zero density or the relative flow is sonic). They are additionally genuinely nonlinear when r1 = 0 or pi or r2 = 0 or pi, or |w| = +-|v|. Dispersive shock waves in a system with these properties have not been studied previously, providing additional motivation and interest in these equations.

5) The authors refer to Landau-Lifshitz theory, which incorporates dissipation. The comparison of the polarization wave equations to magnetism would be more accurate if they referred to it as dissipationless Landau-Lifshitz theory.

6) There are some minor typos, grammatical errors that could be corrected. - p 2, "... species has opened the possibility ..." - p 3, "... components are at rest and both have ..." - p 5, "From this expression we can write ..." - p 20, "... solutions of the celebrated Korteweg-de Vries (KdV) equation ..."

  • validity: top
  • significance: top
  • originality: top
  • clarity: top
  • formatting: excellent
  • grammar: good

Author:  Thibault Congy  on 2016-10-10  [id 62]

(in reply to Report 3 on 2016-09-19)
Category:
answer to question

We thank the referee for his/her eulogistic comments and directly go to the discussion of some of the requested changes. The changes and corrections not mentioned below have been implemented in the iterated version of the paper, we thank the referee for suggesting these ameliorations.

point (2) of the referee. Indeed the comparison of the amplitude of the leading soliton (as obtained via El's method presented in Sec. 5.2) with the numerical result is already displayed in the inset of Fig. 8 (dashed line versus solid line).

point (4): In this paper we confined ourselves to the case when the DSW has a known Gurevich-Pitaevskii structure for demonstration of new remarkable properties of the polarization dynamics decoupled from the
density dynamics. The referee formulates extremely interesting problem for future research. Actually we have hinted at this possibility on page 20 where we write: "In our case the polarization wave dynamics is governed
by the 1D version of Landau-Lifshitz equation that is completely integrable (see, e.g., [32]). However, the Whitham theory is not developed well enough for this equation and therefore El’s method [31] seems the most
appropriate for the description of the DSW observed in Fig. 8." We are now working on the theory of DSWs described by the Whitham equations obtained in Ref.[32] with degenerate points. We completely agree with the referee that "Dispersive shock waves in a system with these properties have not been studied previously, providing additional motivation and interest in these equations."

point(5): We agree with the referee that it is worth mentioning that the system (14) is equivalent to {\em dissipationless} Landau-Lifshitz theory to avoid confusion with real magnetic systems whose accurate description incorporates dissipation effects. We have made such an addition in the revised
version of the paper.

---

## Editorial Decision

published